# Ocean and land forcing of the record-breaking Dust Bowl heatwaves across central United States

Tim Cowan [1,2,3✉], Gabriele C. Hegerl [3], Andrew Schurer [3], Simon F. B. Tett [3], Robert Vautard [4], Pascal Yiou [4], Aglaé Jézéquel [5,6], Friederike E. L. Otto [7], Luke J. Harrington [7] & Benjamin Ng [8]

The severe drought of the 1930s Dust Bowl decade coincided with record-breaking summer heatwaves that contributed to the socio-economic and ecological disaster over North America's Great Plains. It remains unresolved to what extent these exceptional heatwaves, hotter than in historically forced coupled climate model simulations, were forced by sea surface temperatures (SSTs) and exacerbated through human-induced deterioration of land cover. Here we show, using an atmospheric-only model, that anomalously warm North Atlantic SSTs enhance heatwave activity through an association with drier spring conditions resulting from weaker moisture transport. Model devegetation simulations, that represent the wide-spread exposure of bare soil in the 1930s, suggest human activity fueled stronger and more frequent heatwaves through greater evaporative drying in the warmer months. This study highlights the potential for the amplification of naturally occurring extreme events like droughts by vegetation feedbacks to create more extreme heatwaves in a warmer world.

[1] University of Southern Queensland, Toowoomba, Queensland 4350, Australia. [2] Bureau of Meteorology, Melbourne, Victoria 3008, Australia. [3] School of GeoSciences, The Kings Building, University of Edinburgh, Edinburgh EH9 3JW, UK. [4] Laboratoire des Sciences du Climat et de l'Environnement, UMR 8212 CEA-CNRS-UVSQ, IPSL & Université Paris-Saclay, 91191 Gif-sur-Yvette, France. [5] LMD/IPSL, Ecole Normale Superieure, PSL research University, Paris, France. [6] École des Ponts ParisTech, Cité Descartes, 6-8 Avenue Blaise Pascal, 77455 Champs-sur-Marne, France. [7] Environmental Change Institute, University of Oxford, South Parks Road, Oxford OX1 3QY, UK. [8] CSIRO Climate Science Centre and Centre for Southern Hemisphere Oceans Research, Aspendale, Victoria 3195, Australia. ✉email: tim.cowan@bom.gov.au

Many daily maximum and minimum temperature (Tmax, Tmin) records from the 1930s over the continental US still stand as of 2019[1]. These records, like maximum daily Tmax over the central US (Fig. 1a), are unlikely to have resulted from instrumental biases[2,3]. Instead a strong upper-level atmospheric ridge and land–atmosphere interactions may have allowed for extreme heat to build during the Dust Bowl drought[3–6]. The drought, defined through precipitation and evapotranspiration-based indices[3], emerged during a period of cooler-than-average North Pacific sea surface temperatures (SSTs) and a warmer North Atlantic[4,6–10]. Yet when forced with observed SST anomalies, atmospheric-only general circulation models (AGCMs) tend to underestimate the drought's spatial extent and magnitude[4,11,12]. Improved representations of precipitation and temperatures during the Dust Bowl period are simulated when AGCMs[4,11] and regional models[13] implement realistic historical land-cover changes and dust aerosol forcing. Following new insights on the observed extreme heat during the Dust Bowl[3,6], and with future increases in global heatwave activity likely[14], a comprehensive understanding of what contributed to the Dust Bowl heatwaves is crucial.

This study investigates the ocean–atmosphere forcing of the Dust Bowl heatwaves over the central US (105°–85°W, 30°–50°N) in observations, coupled climate models, and AGCMs. Heatwaves are identified when a location's daily Tmax and Tmin is above its respective daily 90th percentile for at least three consecutive days and two consecutive nights, similar to other definitions[15]. Using idealised AGCM simulations, we find that warm Atlantic SST anomalies lead to more frequent Dust Bowl heatwaves over southern–central US than in simulations forced with historical Pacific SST anomalies. This results from a stronger drying in the spring months, stemming from weaker moisture transport from the Gulf of Mexico, leading to a preconditioning of the land surface for extreme summer heatwaves. We then use a set of bare-soil simulations to analyse the extent to which 1930s land-use changes amplified the heat. This reveals the strong sensitivity of heatwaves to increasing bare soil, and supports the hypothesis that the Dust Bowl heatwaves (and partly the drought) were amplified by rapid devegetation and exposed soil from human activities.

## Results

**Role of sea surface temperatures.** The 1930s were the most active summer (June–August) heatwave decade of the twentieth century for the central US with some locations experiencing an average of 22 heatwave days per summer (Fig. 1b; Supplementary Fig. 1), with the longest heatwaves[3] surpassing 10 days in 1934 and 1936, and Tmax anomalies exceeding 6 °C (Supplementary Fig. 1). The record-breaking temperatures and anomalies exceed the spread of responses (in maximum daily Tmax anomalies) from historical experiments from Coupled Model Intercomparison Project Phase 5 (CMIP5) climate models over the 1930s (Fig. 1a).

The underlying decadal SST anomalies in the 1930s resembled the warm phases of the Atlantic Multidecadal Oscillation (AMO)[16,17] and the Pacific Decadal Oscillation (PDO)[18,19] (Fig. 1b). To investigate if similar decadal SST patterns also coincide with heatwave conditions over the central US in unforced model

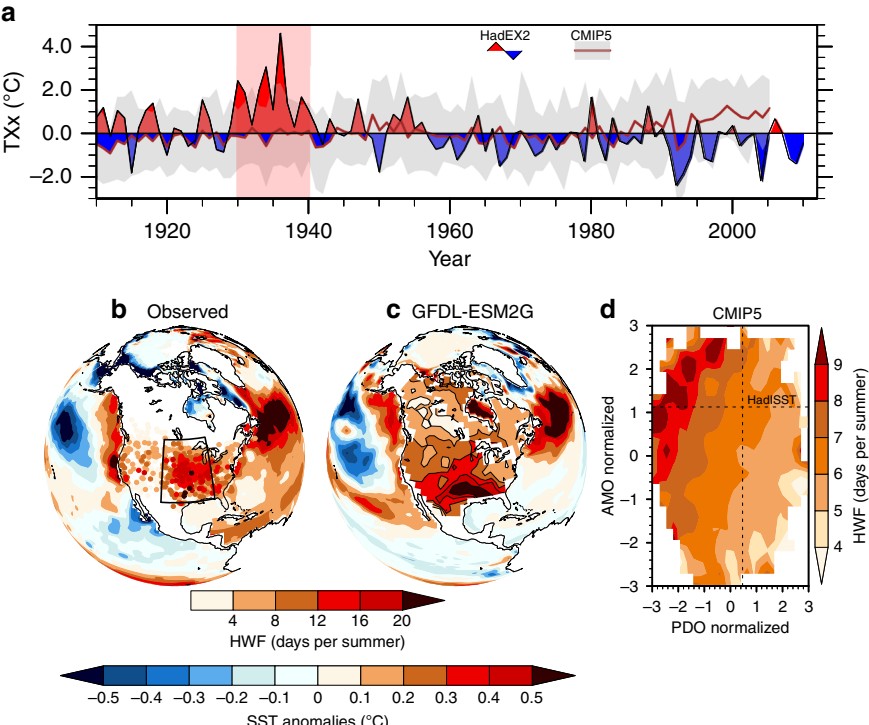

**Fig. 1 Observed and simulated central US summer heatwave activity and associated sea surface temperatures. a** Observed summer (June–August) maximum daily maximum temperature (TXx) anomalies (relative to 1901–2010) averaged over the central US (box in **b**), from gridded observations (HadEX2; colour), and 20 Coupled Model Intercomparison Project Phase 5 (CMIP5) historical model simulations (grey line with shading indicating the 10–90th percentile range; relative to 1901–2005). **b, c** Summer sea surface temperature anomaly and heatwave frequency (HWF) patterns averaged across the most active heatwave summer decade (11 years) per century over the central US for station observations (**b**; pink shade in **a**), and a selected CMIP5 model (Geophysical Fluid Dynamics Laboratory Earth System Model with Generalised Ocean Layer Dynamics component, GFDL-ESM2G) decade with heatwave activity commensurate with observations (**c**). **d** Composite anomalies of central US summer HWF as a function of the normalised annual Pacific Decadal Oscillation (PDO) and Atlantic Multidecadal Oscillation (AMO) indices, from 22 CMIP5 pre-industrial (pi) Control experiments totalling 10,900 years (1930s decadal observations shown as the intersect of dotted lines). The PDO and AMO definition are described in the 'Methods'.

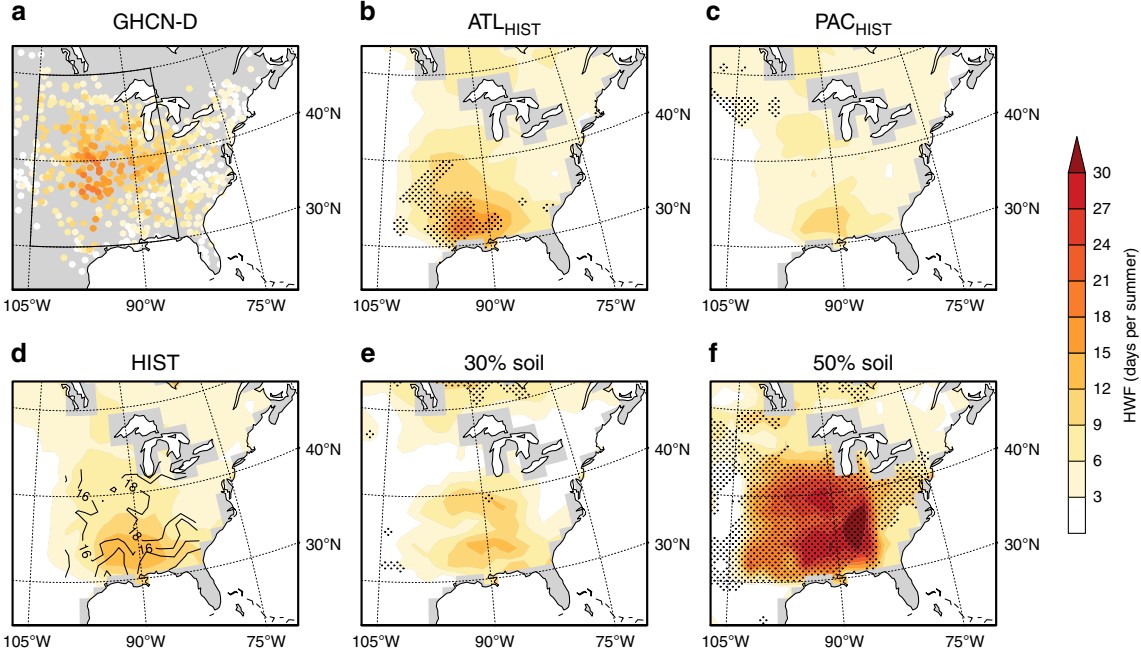

**Fig. 2 Observed and simulated summer heatwave frequency during the Dust Bowl.** Average heatwave frequency (HWF) over the central US for 1930–1937, for observations from Global Historical Climatology Network-Daily (GHCN-D; **a**), and Hadley Centre Global Environment Model version 3 (HadGEM3) simulations, including a five-member Atlantic sea surface temperature (SST) ensemble (ATL$_{HIST}$; **b**), five-member Pacific SST ensemble (PAC$_{HIST}$; **c**), ten-member historical SST ensemble (HIST; **d**), and single-model experiments where the amount of bare soil in 1930, averaged over the central US, was increased from 15 to 30% (**e**) and 50% (**f**). Stippling indicates significantly larger HWF values between (**b**, **c**) ATL$_{HIST}$ and PAC$_{HIST}$ ensembles ($N = 40$) at the 95% confidence level (see 'Methods')[28], and in (**e**, **f**) HWF values outside the HIST simulation range. Contours in (**d**) indicate the percentage of bare soil in each grid cell over the most active heatwave region in the HIST, ATL$_{HIST}$ and PAC$_{HIST}$ members. The climatology period is defined as 1916–1955 (**a–d**) and 1916–1940 (**e**, **f**).

simulations, we use one pre-industrial (pi) Control experiment each from 22 CMIP5 models (~500 years per model, see 'Methods' and Supplementary Table 2). When averaged over almost 11,000 model years, a multi-model ensemble (MME) mean of summer heatwave frequency (HWF; total number of heatwave days) shows that simulated central US heatwaves are more frequent during a warm AMO phase and a cool PDO phase (Fig. 1d). While the simulated PDO phases for higher HWFs do not match with 1930s observed state, heat extremes[20] and drought periods[18] over central and eastern US in the late 1990s to early 2000s were associated with a similar decadal pattern to the CMIP5 piControl MME. From the nearly 11,000 CMIP5 piControl years, we identify one decade per century per model where record-breaking heatwave activity is commensurate with the 1930s observations. In terms of geographic extent, record-breaking heatwave activity is rare in the piControl simulations, as is the clustering of events in a decade (see 'Methods'). Only one CMIP5 piControl decade, in GFDL-ESM2G (model years 286–296), breaks HWF records over more than 50% of northern and southern–central US, similar to observations (Fig. 1c; Supplementary Table 2). This simulated decade also features a warm north Atlantic and anomalously cooler eastern Pacific (i.e., warm PDO and AMO phases), indicating that rare Dust Bowl-like heatwaves can be simulated in the absence of external forcings like land-cover change and dust. This model is also one of a minority of CMIP5 models that is able to represent AMO-like behaviour in its piControl simulation[21].

AGCM studies have emphasised the importance of Pacific SST anomalies to central US droughts[22,23], even for the 1930s[4,12]. We separate the influence of Atlantic and Pacific SST anomalies on the summer heatwaves across the 1930s using AGCM

experiments conducted in HadGEM3-GA6[24] for the early twentieth century warming period (1916–1955; see 'Methods' for the experiment design). This period yields ~15 years of data on either side of the 1930s in order to create an extended climatology to compare the Dust Bowl years against. We define the Dust Bowl period as 1930–1937 to encompass the summers with the most extreme heatwave conditions observed in the 1930s[3], and the HadGEM3 ensembles that capture atmospheric variability are expected to sample it. We first look at a HadGEM3 ensemble forced with historical SSTs (HIST). The HIST ensemble underestimates heatwave activity (comparing Fig. 2a, d), and generates well-documented biases in temperature[4,7] and heatwave magnitude (average temperature of heatwaves) over the southern–central US (105°–85°W, 30°–40°N; Supplementary Fig. 2). Such mean-state temperature biases and anomalous responses to 1930s SSTs exist in other AGCMs, particularly over central and southern North America (contours, Supplementary Fig. 3). Warm model biases over central and southern North America[25] are thought to be due to overly strong land–atmosphere coupling which can amplify heat even in wet regions[26] and lead to persistent drought over the Great Plains[23]. Focusing on HWF, a metric defined relative to a model's own climatology is a partial solution to this issue; however, biases in HadGEM3's physics could still infiltrate and artificially accelerate heatwave development. This is why improvements in the parameterisation of land-surface models is critical, particularly for reducing uncertainty in future climate model projections[27].

To separate the influence of the Atlantic (ATL$_{HIST}$) and Pacific (PAC$_{HIST}$) SSTs, we carry out simulations in which SSTs outside their respective domain cycle through climatological values. The resulting patterns from the ATL$_{HIST}$ ensemble show a

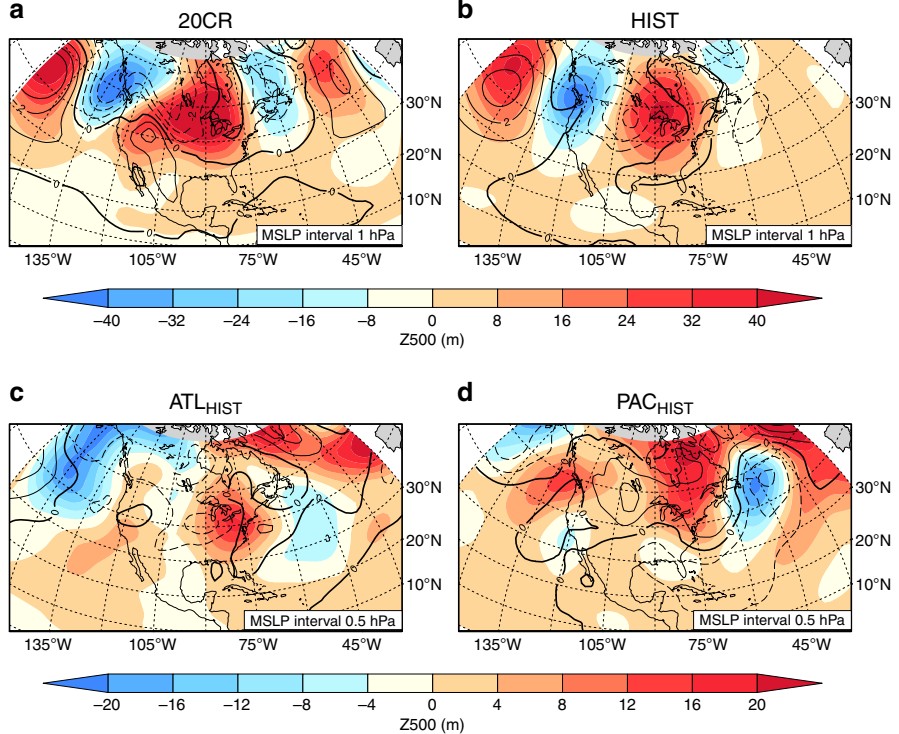

**Fig. 3 Atmospheric conditions during the hottest Dust Bowl heatwaves.** The spatial patterns of mean sea-level pressure (MSLP; colours) and geopotential height at 500 hPa (Z500; contours), averaged over a 7-day period from the start of the hottest summer heatwave over the central US, and averaged over 1930–1937. Shown are the twentieth century reanalysis version 2c (20CR; **a**), a ten-member Hadley Centre Global Environment Model version 3 (HadGEM3) historical sea surface temperature (SST) ensemble mean (HIST; **b**), a five-member Atlantic SST ensemble mean (ATL$_{HIST}$; **c**), and a five-member Pacific SST ensemble mean (PAC$_{HIST}$; **d**). The thick line indicates the 0 hPa MSLP anomaly, while the solid (dashed) lines indicate positive (negative) MSLP anomalies.

significantly stronger response in heatwave activity and intensity across the southern–central US than the PAC$_{HIST}$ ensemble (95% level based on a Mann–Whitney $U$ test[28], Fig. 2b, c; Supplementary Figs. 1c and 2), consistent with the CMIP5 piControl analysis highlighting the importance of Atlantic SSTs (Fig. 1). How continental US surface temperatures respond to remote SSTs varies across AGCMs[22] and coupled models[23], and often depends on a models' middle to upper-level circulation response to SST forcing, particularly over the southern US[5,29].

**Role of summer circulation.** The spatial extent and amplitude of the hottest heatwaves varied considerably during the 1930s[3], yet events often coincided with a persistent high mean sea-level pressure (MSLP) anomaly over the western US (Fig. 3a), coupled with a mid-tropospheric ridge[5,6]. One possibility is that the summer atmospheric circulation response to SST anomalies was the main contributor to the heatwave activity. To test if this explains the stronger ATL$_{HIST}$ heatwave response, we examine the daily atmospheric circulation at the surface (i.e., MSLP) and in the mid-troposphere (i.e., 500 hPa geopotential heights; Z500) during the hottest heatwave weeks for each summer across the simulations (see 'Methods'). For the HIST ensemble, it captures the observed mid-tropospheric ridge across the eastern US associated with the hot conditions over the central US, which is coupled to a surface low anomaly at 100°W, 45°N (Fig. 3b). The heat low and upper-level ridge anomalies are also prominent features in the ATL$_{HIST}$ ensemble (Fig. 3c); however, the surface high anomaly in the western US is absent. In the PAC$_{HIST}$ ensemble, the upper-level ridge is placed further northwards (50°–60°N), while the surface low anomaly is considerably weaker

and displaced southwards over the southeast US coast (Fig. 3d), consistent with a more muted heatwave response. This suggests that for HadGEM3, the main circulation features associated with the hottest Dust Bowl heatwaves across the eastern US are best reproduced with an Atlantic SST forcing. Yet the summer heatwave circulation differences ($\Delta[$ATL$_{HIST}$, PAC$_{HIST}]$) over the central US are not significantly different (based on Kolmogorov–Smirnov two-sample test; also see Supplementary Fig. 4e, f) and cannot fully explain the summer heatwave intensity differences between the ATL$_{HIST}$ and PAC$_{HIST}$ ensembles. We turn to the role of dry springs in the lead up to the Dust Bowl summers.

**Dry spring preconditioning summer heatwaves.** A key factor in observed summer heat extremes over the Great Plains is spring-time preconditioning[6]. Observational studies suggest that dry springs pre-conditioned the Dust Bowl summer heat extremes[3,6,10], driven, in part, by mid-tropospheric ridging[10] and reduced moisture advection from the Gulf of Mexico[9]. Prior to summer, significant April–May precipitation deficits emerge throughout the southern–central US in the ATL$_{HIST}$ ensemble compared to PAC$_{HIST}$ (Fig. 4a). The largest precipitation deficit for ATL$_{HIST}$ occurs mid-May (Supplementary Fig. 4a) and corresponds to comparatively lower evaporative fractions, higher daily Tmax (although no difference in Tmin[30]) and a deeper surface low throughout the summer months (Supplementary Fig. 4b, c, e). The drier and hotter ATL$_{HIST}$ ensemble conditions also partially reflect the weaker meridional moisture fluxes from the Gulf of Mexico in April–May (Fig. 4a; Supplementary Fig. 4d), reminiscent of the observed 850 hPa wind anomalies in

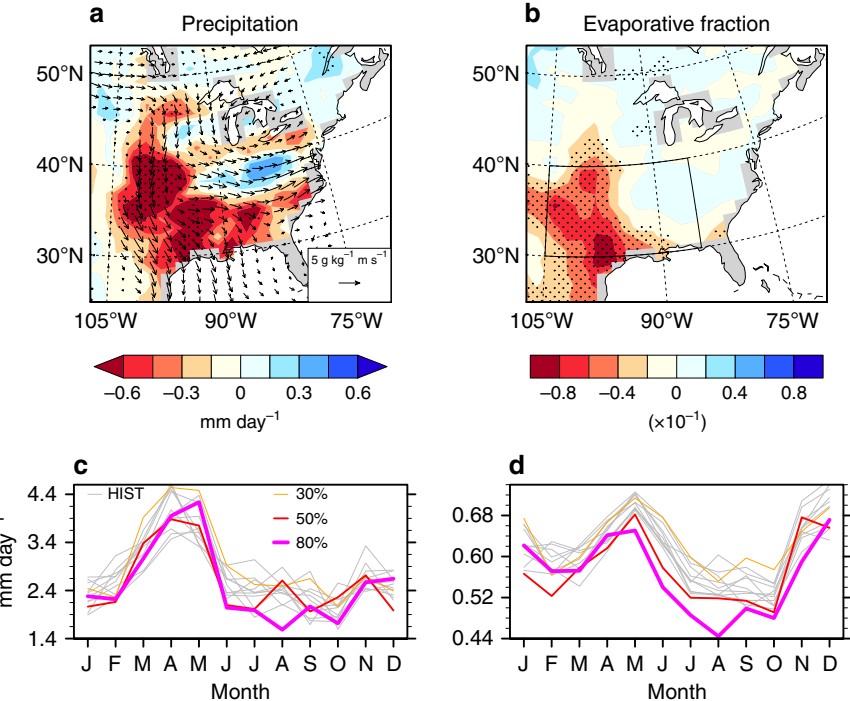

**Fig. 4 Late spring conditions prior to the Dust Bowl summers. a, b,** Composite differences in April–May conditions between the five-member Atlantic sea surface temperature (SST) ensemble mean (ATL$_{HIST}$) and five-member Pacific SST ensemble mean (PAC$_{HIST}$) prior to the Dust Bowl summers for 1930–37. The fields shown are precipitation (colours; **a**) and 850 hPa moisture flux (vectors; **a**) and evaporative fraction (**b**). Stippling on precipitation and evaporative fraction contours indicates significance above the 95% confidence level based on a bootstrap mean difference on 10,000 × 8 April–May samples (resampled). **c, d** Annual cycle of precipitation (**c**) and evaporative fraction (**d**) for the ten-member historical SST members (HIST) and bare-soil model experiments averaged over 1930–1937 and the southern–central US (region in (**b**); 105°–85°W, 30°–40°N).

spring during the mid-1930s[6]. While it appears difficult to distinguish the absolute mid-tropospheric circulation (e.g., Z500) over the eastern US between the idealised SST HadGEM3 ensembles (Supplementary Fig. 4f), the ATL$_{HIST}$ ensemble Z500 is consistently higher than the PAC$_{HIST}$ ensemble by 5–15 m over March–May. Hence, warm Atlantic SST anomalies are more conducive to forcing stronger mid-tropospheric ridging in spring during the 1930s. This weakens the northward moisture fluxes, driving a precipitation deficit and a drying which extends to summer, forcing more active and intense heatwave conditions[3,6].

It is well established that systematic regional precipitation biases exist in AGCMs simulating the Dust Bowl drought[4,7,8,12]. This is true of HadGEM3 and three other AGCMs that have conducted 1930s historical SST-forced experiments and simulate wet spring biases (Supplementary Fig. 5). Even with HadGEM3's wet bias, ATL$_{HIST}$ exhibits comparatively stronger precipitation deficits in mid-spring to summer (than PAC$_{HIST}$), and lower evaporative fractions, consistent with an overly responsive land–atmosphere coupling, in line with other models[26,31]. The majority of forecast systems also overestimate the relationship between dry late springs and summer heat biases due to an excessive reduction in soil moisture[30]. The warm bias in HadGEM3 further causes overly deep surface low anomalies from July (Supplementary Fig. 4e), which exacerbate the average Dust Bowl heatwave temperatures (Supplementary Fig. 2). In overestimating the land–atmosphere coupling strength, HadGEM3 dries out too rapidly in late spring, yet processes controlling the heatwave development can still be compared between differently forced simulations[10,22,29]. Other important processes, such as land-cover changes[4,13] in the 1930s, are examined next.

**Role of land-cover changes.** Although the role of SSTs in triggering the Dust Bowl Drought is recognised[11], land-cover changes likely amplified the heatwaves[13]. Land-cover changes across the US Great Plains in the 1920s and 1930s included widespread crop failures surpassing 60% in many counties[32]. Estimates of the amount of bare soil range from 20 to 80%, depending on regions with or without erosion[13], with AGCMs likely underestimating the true amount[4]. In order to evaluate the contribution by land-cover changes, we conducted sensitivity experiments using HadGEM3, in which temperate C3 and tropical C4 grass cover over the central US-wide was converted to bare soil. This represents the repeated crop failure[4,32] during the 1930s that led to soil exposure and expansive and rapid loss of top soil[10]. In three separate model experiments, the percentage of bare soil averaged over the central US in 1930 was increased from an approximate 15% reference fractional extent to 30%, 50 and 80%, at the expense of C3 and C4 grass. This represents a percentage loss of grass, averaged over the central US in 1930, of 25%, 50% and 100% (Supplementary Fig. 6), which is restored to HIST fractions by 1940 (see 'Methods' and Supplementary Fig. 7). As a comparison, Cook et al.[4] used a conservative estimate of up to 50% devegetation in some grid boxes in their 1930s simulations. Our experiments reveal a significant increase in HWF from around 6 days in the 30% soil experiment (Fig. 2e) to ~17 days in the 50% run (Fig. 2f), eventually reaching an average of 33 days per summer in the 80% run over the southern–central US (Supplementary Fig. 2). With more exposed bare soil, an earlier evaporation deficit appears to drive the elevated heatwave response through soil desiccation, despite only minor differences in precipitation between the 50% and 80% soil experiments (Fig. 4c, d). Reductions in early summer soil moisture and latent heat fluxes

provide further evidence of an earlier drought emergence under increasing crop removal, leading to a thicker atmospheric boundary layer (Supplementary Fig. 8e–g). A more dramatic HWF response in the heavily forested areas to the south[33] than the grasslands to the north and west, stems from the smaller diurnal temperature variations in the tropical latitudes. This has a larger effect on HWF as the heatwave threshold is easier to exceed, and hence a small increase in summer temperatures can lead to a larger HWF signal than over a region with more variable summer temperatures[14]. Part of bare-soil induced heatwave impacts stem from greater temperature advection along the Gulf regions, although this occurs mainly along the coast. High-resolution modelling experiments suggests the primary role that land-cover changes had in amplifying the Dust Bowl drought was by reducing the moisture transport to the central US[13]; however, we find little appreciable change in moisture fluxes (Supplementary Fig. 8b). Other studies have shown elevated 1930s dust aerosols[4,11] would have suppressed precipitation through land–atmosphere feedbacks although these studies excluded the indirect aerosol effect on cloud microphysics. We see no evidence of land albedo differences in our bare-soil experiments (Supplementary Fig. 8h), despite the high dust loadings (Supplementary Fig. 6), although it is possible that precipitation efficiency is reduced via the dust-radiative effects on clouds[34]. Our results suggest that regional-scale land-surface processes play a more dominant role over the large-scale atmospheric dynamics and land–atmosphere feedbacks in our bare-soil experiments. If HadGEM3 had a more accurate spatial and temporal representation of the 1930s land cover[13] that distinguished the less drought-resilient dryland crops (due to their shallower root system), this may have produced more realistic heatwaves, yet it would still not capture important land–atmosphere feedbacks that result from increased soil erosion and elevated dust levels[4,11].

## Discussion

Supported by the HadGEM3 results (Fig. 2) and consistent with unforced CMIP5 simulations (Fig. 1), this study reveals that Atlantic SSTs were an important factor in the Dust Bowl heatwaves, enhancing spring drought, and allowing heat to develop earlier over the central US[3,6]. Pacific SSTs played less of a role in the heatwave development, at least in our AGCM simulations (Fig. 2), but have been shown to contribute to the Dust Bowl drought in other AGCMs[7,8] and more generally, to long-term droughts over the Great Plains in coupled models[23]. Yet AGCMs forced with observed SSTs alone or historical boundary conditions remain far from quantitatively reproducing the exceptional nature of the Dust Bowl heatwaves (or drought) (e.g., Fig. 2). This is likely stems from how models treat 1930s land-cover changes, resulting in rapid deep plowing of native grassland. Subsequent drought brought about widespread crop failures[32,35], dust storms[36] and protracted heatwaves[3]. Despite the uncertainty behind the exact amount of devegetation prior to and during the 1930s[4,32] (see Soil Conservation Service erosion map[37]), our grass devegetation experiments show a strong enhancement in heatwave activity and intensity (i.e., absolute maximum temperatures) as bare-soil amounts increase. This is due to a greater partitioning of surface energy fluxes to sensible heat and a rapid drying of exposed soil (Fig. 4), although elevated dust loads appear not to suppress precipitation in HadGEM3 as in other AGCMs[11]. The 1930s extreme heatwaves serve as a reminder that vegetation feedbacks from a greenhouse gas-induced warming, resulting from human activities, could lead to the enhancement of extreme heatwaves and drought triggered by decadal climate variability. Even with recent crop intensification linked to cooler and wetter conditions over the central US[38], major crop losses would likely

still occur if future Dust Bowl-like heatwaves eventuate[35,39] due to a warmer world.

## Methods

**Heatwave metrics and surface heat fluxes**. The use of consecutive hot days and nights (3 days for Tmax and 2 nights for Tmin) above their respective daily 90th percentile threshold is similar to other heatwave definitions[15]. The threshold is calculated over the full climatological period, using a centred 15-day window that removes monthly and seasonal dependencies. Heatwave metrics are then separated into two categories: (1) activity; total number of summer heatwave days (frequency; HWF), and longest summer heatwave event (duration; HWD); and (2) intensity; average temperature of all summer heatwaves (magnitude; HWM). As we do not study the human health impact, we do not consider relative humidity (increasing the heat stress[40]) in our heatwave definition. Even with the lack of reliable humidity observations in the 1930s[41], the study's main focus is on the record heat, and its association with SSTs and bare soil. We acknowledge HadGEM3 contains a strong warm bias in summer hot extremes and a wet spring bias over North America, along with other AGCMs (see Supplementary Figs. 2, 5), most likely due to an over-active land–atmosphere coupling[26]. As such, we primarily focus on HWF as this metric partially accounts for warm model biases, as temperatures are referenced to a model's own climatology.

Evaporative fraction (EF) is used as a proxy for soil moisture, calculated from latent ($Q_e$) and sensible ($Q_h$) heat fluxes as $EF = Q_e/(Q_e + Q_h)$. EF is the ratio of incoming energy used for evapotranspiration to the total amount of incoming energy. A more arid surface typically has lower EF values and this reduces evaporative cooling[42]; however, the interactions between soil moisture and EF can be modulated by daily net radiation and meteorological conditions[43].

**Observational and reanalysis data**. Daily temperature observations are from the Global Historical Climatology Network-Daily (GHCN-D) archive[44]. Further details on the extensive coverage in the 1930s and the station selection can be found in Cowan et al.[3]. Maximum daily Tmax anomalies in Fig. 1a are taken from the gridded HadEX2 dataset[6] (for a HWF-based version of Fig. 1a, see Supplementary Fig. 1a). Our hottest heatwave dates were determined from temperatures obtained from the Berkeley Earth Surface Temperature (BEST) dataset[45], which incorporate the large network of GHCN-D stations (Supplementary Fig. 2a shows a HWM comparison between BEST and GHCN-D). Given the daily BEST product is considered experimental, we predominantly focus on GHCN-D for the spatial representation of the observed heatwaves. Daily mean sea-level pressure (MSLP) and 500 hPa geopotential height (Z500) fields are from the US National Oceanic and Atmospheric Administration's (NOAA) Twentieth Century Reanalysis (20CR) version 2c[46]; the reanalysis product assimilates daily observations of surface pressure with monthly SSTs and sea ice as boundary conditions from Hadley Centre Sea-Ice and Sea-Surface Temperature Data Set Version 1 (HadISST1). Monthly precipitation is taken from both GHCN-D and Global Precipitation Climatology Centre (GPCC)[47], as precipitation (and surface heat fluxes) from 20CR contain an abundance of artificial inhomogeneities over the central US prior to the 1950s arising from changes in observational density[48].

**HadGEM3-GA6 SST-forced experiments**. The SST-forced atmospheric model experiments are generated using the HadGEM3-GA6 model with a N96 (~210 km at equator, 135 km at 40°N) horizontal resolution (1.25° × 1.875° and 38 vertical levels in the atmosphere). It is based on the Met Office Unified Model Global Atmosphere 6.0 (Version 8.5) and Joint UK Land Environment Simulator (JULES) land-surface model 6.0[24]. HadGEM3 is driven by observed daily SSTs, interpolated from monthly values from HadISST2.1, which contains a more comprehensive coverage of in situ observations and more complete bias corrections than HadISST1[17].

The HadGEM3 experimental set-up comprises of ten historical simulations (HIST) run over 1916–1955 with each member forced with a different SST realisation that encompasses observational uncertainties and bias adjustments[49] and standard CMIP5 historical drivers (e.g., well-mixed greenhouse gases, aerosols, land-use changes). A further two sets of five idealised simulations were conducted: the first set kept Atlantic SST anomalies at historical observed values (ATL$_{HIST}$), while the second set kept Pacific SST anomalies at historical observed values (PAC$_{HIST}$). In both ensembles, the remaining oceans cycled through a 1916–1955 climatology for each year, with anomalies joining at 20°S and 60°N, far from the region of interest. The SST and ice regions below and above the aforementioned latitudes are kept at climatology in both experiment ensembles. The residual response was computed as the difference between the ATL$_{HIST}$ and PAC$_{HIST}$ ensembles following the approach of Schubert et al.[7]. All model-based heatwave metrics are calculated relative to the ten-member HIST ensemble climatology from 1916–1955. HadGEM3's HIST mean-state precipitation and surface temperature are within the range of three other AGCMs that have conduced similar SST-forced experiments (Supplementary Figs. 3, 5). The three comparison AGCMs are GFDL-AM3 (~1.9° × 1.9° horizontal resolution), GEOS-5 (1.25° × 1° resolution) and ESRL-CAM5 (0.5° × 0.5° resolution). The warm bias in over the southern–central US in HadGEM3 is accentuated in the soil experiments with maximum

temperatures ranging from 43 °C to 52 °C (Supplementary Fig. 8a), compared with observed temperatures of 32–34 °C.

**HadGEM3-GA6 bare-soil experiments.** Each land grid cell consists of different fractions of surface types, like grass, bare soil, trees and shrubs. Devegetation experiments were generated by converting C3 (temperate) and C4 (tropical) grassland fractions over the central US to bare soil. The land fraction ancillary files are linearly interpolated between decades, meaning values are given every decade (e.g., 1920, 1930, 1940). The soil experiments are labelled 30%, 50% and 80% soil, which refers to the approximate percentage of bare soil in 1930 averaged over the central US (base soil percentage in HIST is 15.3%). The equivalent combined C3 and C4 grass loss, averaged over the central US at 1930 for the three experiments, is ~25%, 50% and 100% (Supplementary Fig. 6). This subsequently increases the dust aerosol load in the spring season over the same region. In each of these experiments, the bare-soil percentage returns to 15.3% by 1940, meaning the temporal rate of grass *recovery* differs between the three experiments during the 1930s decade (Supplementary Fig. 7). For comparison, Cook et al.[4] use up to a 50% vegetation loss over particular grid points in similar experiments in the GISS ModelE over the central US Great Plains (105°–95°W, 30°–50°N), which they assume to be a conservative estimate due to the exclusion of natural vegetation loss during the Dust Bowl drought. Only one soil sensitivity experiment is conducted per soil fraction, and as such, given the lack of multiple ensemble members, statistical significance is based on whether a grid point lies outside the ten-member HIST range. For example, the HIST mean summer mean surface temperature for the southern Great Plains over 1930–1937 ranges from 30.2 °C to 31.3 °C across the ensemble members, whereas the 50% and 80% soil experiment temperatures of 32.8 °C and 35.5 °C, respectively, exceed the HIST range, and therefore are deemed to be statistically significant. As the bare-soil sensitivity experiments are rudimentary in their design, the aim was not to realistically quantify their imprint on the Dust Bowl heatwaves, but to gauge the impact devegetation can have on the amplification of heatwaves and to test for land-atmospheric feedbacks. As with previous studies using land-cover estimates from the 1930s[4,13], the assumptions of bare soil amounts remain subjective.

**Circulation patterns.** For the reanalysis and each model simulation, we find the dates of the hottest summer heatwave week over the central US. To achieve this, for each grid cell and each summer, we first calculated the heatwave start date using the heatwave amplitude (i.e., the hottest day of the hottest heatwave), and then determined how many grid cells shared the same date. A 7-day running mean was performed over all start dates to choose the week with the largest percentage of central US grid cells that shared the same heatwave start date (centred in the week). We then average each simulation's MSLP and Z500 over the hottest heatwave week and over 1930–1937 to create the composites shown in Fig. 3. Each composite is made up of 7 days × 8 years × 5 (or 10) experiments, leading to 280 patterns for ATL$_{HIST}$, PAC$_{HIST}$ and 560 patterns for HIST.

**CMIP5 experiments.** Heatwave metrics, calculated from daily Tmax and Tmin, and monthly SSTs were analysed from the piControl experiments of 22 CMIP5 models[50] (Supplementary Tables 1, 2). All model data were interpolated onto a regular 1° grid prior to analysis. Models with at least 400 piControl years of output were considered (all except for bcc-csm1-1-m have 500 years), which allowed for an assessment of multiple occurrences of record-breaking heatwaves over at least four centuries for each individual model piControl experiment. To ensure we captured decadally clustered events as observed during the Dust Bowl decade, rather than outlier record-breaking heatwaves, for each models' HWF, we applied an 11-year (summer) running average over each central US grid point for the entire piControl period. For each century (e.g., years 1–100, years 101–200, to years 401–500), we selected the maximum HWF value for each grid point and noted the decade of occurrence. We then determined the percentage of central US grid points where record-breaking values fell into the same decade, with the highest percentage selected as the record-breaking decade for that century for that particular model. If two record-breaking heatwave decades occurred within a 30-year time span (e.g., end of one century and beginning of the next), the second decade was ignored to reduce possible contamination from the same decadal SST pattern. This process produced 103 individual piControl heatwave decades. The years when record-breaking heatwaves occurred based on HWF and centred on the decade in question, and the percentage area of the central US impacted, are shown for each CMIP5 model in Supplementary Table 2. The GFDL-ESM2G model is highlighted in Fig. 1 as an example of one such model decade that realistically matches the observations in terms of its SST patterns and heatwave activity (e.g., it breaks similar HWF records over a large spatial extent of the central US, as seen in the 1930s). To test if the simulated heatwave peaks are actually decadal in nature and not random occurrences, we conducted bootstrapping analysis on the CMIP5 piControl runs (10,000 times with replacement) for clusters of at least four significant heatwave years in a decade that surpass one-standard deviation above the long-term mean. For a heatwave metric like duration, of the 21 CMIP5 models with 500 piControl years, 11 models exhibit decadal cluster frequencies above the bootstrapped third quartile, with six at or above the 95th percentile. This infers that

substantial decadal clustering similar to the 1930s, occurs in about half the models, and strongly in less than 30% of models. These results are similar using heatwave amplitude.

For the historical experiments, we selected 20 CMIP5 models (first run only) to assess the heatwave behaviour and maximum daily Tmax in the historical period simulations (Fig. 1a; Supplementary Fig. 1a). These include the 17 CMIP5 models listed in Supplementary Table 2, as well as CCSM4, IPSL-CM5A-MR and GFDL-CM3 (these three models did not have piControl experiments available). The multi-model ensemble mean shows the forced response on HWF (representing heatwave activity) and HWM anomalies (representing heatwave intensity). We would expect to see an increase in the 1930s as there was a large enough forced response in temperature extremes to land-cover changes (i.e., signal to noise), however CMIP5 models are unable to capture these extremes as represented by maximum daily Tmax anomalies (Fig. 1a). An increase in irrigation, not included in CMIP5 models[51], through intense agriculture over the eastern Great Plains is thought to be related to a lack of a trend in summer temperatures[52], compared to CMIP5 models since the 1950s (Supplementary Fig. 1a).

The Pacific Decadal Oscillation (PDO) and Atlantic Multidecadal Oscillation (AMO) annually averaged time series were calculated for the same piControl period as the heatwave metrics using NCAR's Climate Variability Diagnostics Package Repository. The PDO is defined as the leading Principle Component of North Pacific (20–70°N, 110°E–100°W) area-weighted SST anomalies (with global mean SST anomalies removed). The AMO is defined as area-weighted SST anomalies averaged over the North Atlantic (0–60°N, 80°W–0°E), with the global (60°S–60°N) mean SST anomaly removed. No low-pass filtering is performed on the decadal oscillation indices. We included all available CMIP5 models in our multidecadal analysis, although we are aware that the majority of models lack internally generated Atlantic multidecadal variability[53].

**Significance tests.** In Fig. 2, the non-parametric two-tailed Mann–Whitney $U$ test[28] is used to test the significance in heatwave differences between the ATL$_{HIST}$ and PAC$_{HIST}$ ensembles for each grid point. The null hypothesis tested here is that the data from ATL$_{HIST}$ and PAC$_{HIST}$ have been drawn from the same distribution. The Mann–Whitney $U$ test determines whether the experiment in question is distinguishable from its partner experiment at the 95% confidence level. The Mann–Whitney $U$ test is conducted on the 1930–1937 heatwave summers metrics, per ensemble member, totalling 80 separate cases (8 events per member ×5 experiment members ×2 experiments). Significance for the precipitation and evaporative fraction (Fig. 4) is tested using a bootstrapping method[54], whereby HadGEM3 experiment members for either ATL$_{HIST}$ or PAC$_{HIST}$ are concatenated to form 200 samples of the same month, made up of 5 members ×40 years (1916–1955). Differences between two months from the 200 month sample are randomly resampled 10,000 times and significance are detected if the Δ(ATL$_{HIST}$, PAC$_{HIST}$) residual lies outside 2.5–97.5% confidence bounds of the resampled distribution. The significance is unaffected if choosing anomalies or the mean states for the difference calculation.

## Data availability

The 20CR circulation variables of Z500 and MSLP can be downloaded at https://www.esrl.noaa.gov/psd/data/gridded/data.20thC_ReanV2c.html. Daily temperature and precipitation data are from the GHCN-Daily archive are available from http://www1.ncdc.noaa.gov/pub/data/ghcn/daily. Daily SSTs from HadISST2.1 are available on request. Monthly precipitation from GPCC are available from https://www.esrl.noaa.gov/psd/data/gridded/data.gpcc.html. The HadGEM3 experiments are available on request. The CMIP5 piControl experiments are available via https://esgf-node.llnl.gov/projects/cmip5, and can be downloaded with a valid Earth Systems Grid account. The AMO and PDO time series from the CMIP5 piControl simulations can be downloaded from NCAR's Climate Variability Diagnostics Package Repository: http://www.cesm.ucar.edu/working_groups/CVC/cvdp/data-repository.html. The comparison AGCMs can be obtained from https://psl.noaa.gov/ courtesy of the NOAA-ESRL Physical Sciences Laboratory.

## Code availability

The code to generate the figures is available at https://github.com/tcowan80/Cowan_et_al_2020_DustBowl_HadGEM3, and is based on the NCAR Command Language (version 6.6.2; doi:10.5065/D6WD3XH5). The code to generate climatologies was constructed using Climate Data Operators (CDO) version 1.9.8.

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

## Acknowledgements

This study forms part of the Transition into the Anthropocene (TITAN) project, funded by a European Research Council (ERC) Advanced Grant (EC-320691), and supported by the EUCLEIA project funded by the European Union's Seventh Framework Programme (FP7/2007–13) under Grant Agreement 607085 and the EUPHEME ERA4CS grant 690462. T.C. received additional support from the Northern Australian Climate Program, with funding provided by Meat and Livestock Australia, the Queensland Government and University of Southern Queensland. G.C.H. was also supported by National Centre for Atmospheric Science (NCAS) and as a Royal Society Wolfson Research Merit Award (WM130060) holder. A. S. was supported by Belmont/JPI-Climate Project PACMEDY (Paleo-Constraints on Monsoon Evolution and Dynamics; NERC: NE/P006752/1). P.Y. was supported by an ERC Grant 338965—A2C2. The HadGEM3 model experiments for this project were conducted through the UK's National Supercomputing Service ARCHER. The authors acknowledge Mike Mineter for his programming assistance.

## Author contributions

T.C. and G.C.H. designed and wrote the study. S.F.B.T. and A.S. assisted with conducting the HadGEM3-GA6 simulations. F.E.O. and L.J.H. provided additional modelling support. P.U. and A.J. supplied a framework to evaluate the atmospheric circulation. R.V. and B.N. helped interpret the results. All authors assisted with the analysis and review of the paper versions.

## Competing interests

The authors declare no competing interests.
