## [Peer Review File · Nature Communications]

Reviewers' comments first round:

Reviewer #1 (Remarks to the Author):

General Comments:

The takeaways I had were that (1) atmospheric circulation alone (through pattern matching of Z500) inadequately explained the magnitude of heat extremes, (2) teleconnections manifest through SST forcing from the Atlantic had a stronger influence than that from the Pacific in the Great Plains drought signal, (3) antecedent soil drying due to subpar spring precipitation preconditions extreme temperatures and that this signal is also primarily from the Atlantic basin, and (4) vegetation degradation exacerbates temperature extremes. This is a logical set of results that build on much prior work on this topic. This work is quite strong from a statistical perspective, however, the atmospheric dynamics are not well articulated in its current form. I have provided a couple of main areas where the work can be improved upon.

Major Considerations

1. The climate models used have well-known deficiencies in simulating multidecadal variability which likely needs to be emphasized for the pre-industrial CMIP5 comparisons. Previous studies have done work at establishing model credibility of low-frequency variability and these may be used to cull from the broader set of models

a) Do models actually show that heat extremes are clustered decadal? Is this the basis for selecting results from GFDL-ESM2G?

b) Lines 305-308 suggest that querying GCMs for HWF records clustered in a 10-year period is considered. Are there any key properties about GCMs that do this different than others and how they capture elements of decadal oceanic variability?

2) The section on atmospheric circulation influences on spring and summer is challenging to follow and not very convincing.

a) Fig S6 shows some evidence of below-normal precipitation during Apr-May for ATLh, although more egregious are the larger positive anomalies from the PACH which may be further emphasized in the results. I would suggest to not alter the magnitude of results (i.e., Observed patterns are reduced by a factor of 0.5). Is there any way to mask out signals that are insignificant (e.g., where the ensemble mean is not different from 0 w/confidence intervals?).

b) Fig S7 and accompanying text (lines 140-155) are hard to follow and don't seem to be well supported by the data visualized. I see very weak spatial pattern correlation between ATLh and the Observed fields. Perhaps calculating this and reporting the values will help overcome what my eyes are drawn to. Notably, the persistent ridging across the West from Mar-June in observations collocated with anomalously high MSLP would deter the low-level influx of moisture from the Gulf of Mexico into the Great Plains which is key to spring-summer water vapor transport. It might help to quantify integrated water vapor fluxes with these simulations (Fig S3e has some of this, but it is reported for region that doesn't totally correspond to the drought area. This could be slightly reworked.

c) I understand the motivation for trying to parse out the contributions from each basin. But, it would be worthwhile considering an experiment that shows the full HIST simulation to potentially assess whether there is some interacting circulation pattern.

Minor considerations

1. Line 25, remove "pre-industrial"

2. Line 37, "prolonged unusual synoptic conditions" is this well understood as being part of a blocking pattern through which land-surface interactions strengthen a large wave pattern?

3. Line 82, Why chose 1930-1937?

4. Tests to examine whether circulation only explain the magnitude of the heat extremes were a bit cumbersome to follow. A few questions follow:

a) It is unclear whether analogs are selected using individual days or individual weeks. The latter

seems more appropriate given the influence of land-surface memory on realized surface temperatures (e.g., short-term rapid drying with heat stress may further enhance Bowen ratio);

b) Are 500hPa heights or 500hPa height anomalies used? The prior might be better, but would likely bias results for models that have mean-state biases in Z500. The latter might correct for mean state anomalies, but have other problems.

c) I do not completely follow the justification for using the Mahalanobis distance. The Mahalanobis distance would be advantageous for multivariate analogs (e.g., joint consideration of MSLP and Z500).

5. Line 158, remove "mid-west", also subsequent line is it fair to said anything here about cropland failures in coupled GCMs - GCMs are not explicitly simulating crop yields or failures, right?

6. Line 162, Are these prescribed bare soil experiments dynamic such that soil can be barren in 1930 and recover through dynamic vegetation processes, or will bare soil in 1930 be bare in 1936?

7. Line 298, Repeat sentence.

8. Line 184, Notably as well to changes in cropland practices and potentially the lack of recent extremes rivaling the 1930s is increased crop intensification and its corresponding role in increasing ET (Mueller et al., 2016). This might be thought of as sort of an opposing experiment to the bare soil approach used here.

9. Fig 3b, The units of Mahalanobis distance are not useful. Consider reporting these as standardized units instead.

Reviewer #2 (Remarks to the Author):

This paper discuss the relative contribution SST forcing, atmospheric circulation and land cover changes on the 1930 dust bowl. The study is interesting, evidence are convincing, the sensitivity experiment set-up is complete. I think it is an interesting study. Despite of this, I don't think this study fits in Nature communication, especially for a question of format. My feeling is that the text is too dense and somehow hard to follow, I think this large and complete study would fit much better in a longer format, such as Journal of Climate. If the author prefers to publish in Nature Communication, I would suggest them to think a bit more carefully which figure and text should be part of supplementary material. Bellow my detailed comments:

Scientific:

The model biases and set-up limitation are not discuss comprehensively in the text, some biases are mentioned, but I miss a paragraph clearly explaining the biases of the HadGEM3: wet spring bias, misrepresentation of the heat low, land atmosphere coupling, warm bias, too weak temperature anomalies, and a clear discussion of how it could affects the results of the present study

I would suggest the author to include Ardilouze et al. (2017), which have discussed the relation between soil moisture and heat wave in the US great plain and how biases in soil-atmosphere coupling lead to misrepresentation of heat wave.

Role of land cover changes: The set-up described here is pretty simple and drastic, how realistic is this set-up? The limitation of the sensitivity experiments should be more carefully discussed.

L180-181: From the text it is not clear to me what make you conclude the land cover change is the most important contributor, further sensitivity experiments, with change imposed SST and change in land cover would be needed to determine the respective contribution of each.

L187-190: I don't understand which result support this conclusion.

Structure:

Role of spring and summer circulation: This paragraph is extremely dense, here I clearly miss in the main text, the visualization of the analog pattern of ATLHist and PACHist. Figure S4, could be replace by the ensemble mean of Z500 analog pattern for observations, and the 2 sensitivity

experiments, and I think this figure should be part of the main text to help the reader. In addition, figure 3b and c shows significant differences in the Mahalanobis distance for 2 years during summer, which is not commented in the text.

From L140 to 156: the text is really hard to follow. My feeling is that model biases and misrepresentation of heat low and precipitation in HIST should be discussed separately from the results of sensitivity experiments.

Minor:

L 140: I understand that "spring" is missing here.

L156-157: Missing transition

L86: could this be related to the bias described in Ardilouze et al. 2017

Reviewer #3 (Remarks to the Author):

Review of Ocean and land forcing of the record-breaking Dust Bowl heat waves across central United States.

This is an interesting and important topic especially given the critical relevance of drought, heatwaves, and our need to understand the interactions and cascading impacts between such processes. Overall the manuscript is well written and well communicated. However, there are a number of questions and clarifications the authors should address prior to formal publication.

The choice of using the 1916-1955 model climatology is not specifically justified in the manuscript. Further, the use of this window is potentially problematic given a large portion of the study domain was locked in another "decadal" drought during the 1950s. In fact, in portions of the Southern Great Plains (e.g., Oklahoma, Texas, etc.) the overall magnitude of the thermal anomaly(ies) during the JJA period in the 1950s was as large or greater than during the dust bowl period. This may be (in part) why this region of the domain does not show as strong of a heat wave signature during the dustbowl interval. Can the authors provide a more detailed justification for using his temporal window versus another?

In the experiments focused on devegetation, and considering the domain used, large portions contain forested areas versus grassland. How were these areas treated? Why do areas that are heavily forested (e.g., the southeast portion of the domain) see such a dramatic HWF signature during the devegetation experiments versus the grassland ecosystems further to the west (nearly double the total number of "days")?

Perhaps this is just semantics, but drought is never formally defined in the manuscript. Additionally, instead of using "drought" in the Spring precipitation analyses, the authors note "precipitation deficits". So, this leads to the question: Did drought occur during the Spring precipitation period as implied? Additionally, timing (as noted by the authors) of precipitation is critical in the study domain because of the asynchronous nature of precipitation/temperature in the region (see Flanagan et al. 2017). Thus, even "normal" or even above normal precipitation occurring earlier in the year could have a similar net effect in the JJA period. In other words, timing and magnitude of precipitation matter.

Revisiting the vegetation portion of the study, the authors approach the link between heatwaves and terrestrial feedbacks focused on the Dust Bowl by removing the vegetation – for obvious reasons. An alternative experiment would be to simply desiccate the existing vegetation versus replacing with bare soil. Are the results similar? The reason for this approach is that, especially when considering future potential outcomes, the lessons learned from the 1930s in terms of land management have yielded starkly different practices and it is highly unlikely that such bare soil type of conditions would ever re-occur in the central US in the future. However, extreme Spring drought could lead to large-scale vegetation desiccation regardless of human efforts. In this

scenario, would the results be similar?

This study uses a heatwave definition solely based on temperature. Yet, there is increased awareness that the combination of extreme temperature combined with anomalous atmospheric humidity has greater human impacts and are likely more concern into the future. While it may make little difference in the overall results in this case, would a more advanced HWF metric incorporating humidity (apparent temperature) be more valid?

The study also generally skirts the land-atmosphere impacts of the analysis while more robustly discussing the SST/teleconnections. This is likely due to uncertainties in the models (as discussed by the authors) and the discussion is primarily focused on one-way processes whereby dry surfaces yield greater HWF. Yet there are known, strong linkages between the surface and atmosphere in the study domain (e.g., Koster et al. 2004, Guo and Dirmeyer 2013, etc.) and the linkages are stronger during drought events (e.g., Basara and Christian 2018, Wakefield et al. 2019). Further, there is likely a pathway to the feedback that noted in Miralles et al. 2018. Do the results of this study show the full pathways between such features including local humidity and vertical profiles that would inhibit precipitation?

Have the authors explored additional analogs to demonstrate the consistency of the results? One scenario could include the decadal drought of the 1950s previously mentioned. This is especially interesting in that many of the lessons learned from the Dust Bowl were employed before and during the drought with overall less severe terrestrial impacts even during pronounced and excessive heat. Additionally, more recently while decadal drought has not occurred in the region, significant annual-type droughts have including 2011 in the southern part of the domain (consistent with the 1950s drought and extreme heat) and 2012 (similar to the 1930s and also included heatwaves).

Finally, the study domain in the central US is actually quite large and incorporates approximately six regional climate regions defined by Karl and Koss (1984). Would the results hold if the domain size was changed or divided into the different subregions noted by Karl and Koss?

Additional comments:

Why does the western part of Lake Superior experience a dramatic increase in HWF during the devegetation experiments?

Line 233 – an end bracket is needed.

Line 269 – need a “the” between “in” and “observations”.

References:

Basara, J. B., and J. I. Christian, 2018: Seasonal and interannual variability of land–atmosphere coupling across the Southern Great Plains of North America using the North American regional reanalysis. *International Journal of Climatology*, 38, 964–978.

Guo Z, Dirmeyer PA. 2013. Interannual variability of land–atmosphere coupling strength. *J. Hydrometeorol.* 14(5): 1636– 1646.

Flanagan, P. X., J. B. Basara, and X. Xiao, 2017: Long-term analysis of the asynchronicity between temperature and precipitation maxima in the United States Great Plains. *International Journal of Climatology*, 37, 3919–3933.

Karl T R and Koss W J 1984 Regional and National Monthly, Seasonal, and Annual Temperature Weighted by Area, 1895–1983 (Historical Climatology Series 4–3) (National Data Center: Asheville, NC) p 38

Koster RD, Dirmeyer PA, Guo Z, Bonan G, Chan E, Cox P, Gordon CT, Kanae S, Kowalczyk E, Lawrence D, Liu P, Lu C-H, Malyshev S, McAvaney B, Mitchell K, Mocko D, Oki T, Oleson KW, Pitman A, Sud YC, Taylor CM, Versegny D, Vasic R, Xue Y, Yamada T. 2004. Regions of

strong coupling between soil moisture and precipitation. *Science* 305(5687): 1138– 1140.

Miralles DG, Gentile P, Seneviratne SI, Teuling AJ. Land-atmospheric feedbacks during droughts and heatwaves: state of the science and current challenges. *Ann N Y Acad Sci.* 2019;1436(1):19–35. doi:10.1111/nyas.13912

Wakefield, R.A., J.B. Basara, J.C. Furtado, B.G. Illston, C.R. Ferguson, and P.M. Klein, 2019: A Modified Framework for Quantifying Land-Atmosphere Covariability during Hydrometeorological and Soil Wetness Extremes in Oklahoma. *J. Appl. Meteor. Climatol.*, 58, 1465–1483.

Reviewers' comments:

Reviewer #1 (Remarks to the Author):

General Comments:

The takeaways I had were that (1) atmospheric circulation alone (through pattern matching of Z500) inadequately explained the magnitude of heat extremes, (2) teleconnections manifest through SST forcing from the Atlantic had a stronger influence than that from the Pacific in the Great Plains drought signal, (3) antecedent soil drying due to subpar spring precipitation preconditions extreme temperatures and that this signal is also primarily from the Atlantic basin, and (4) vegetation degradation exacerbates temperature extremes. This is a logical set of results that build on much prior work on this topic. This work is quite strong from a statistical perspective, however, the atmospheric dynamics are not well articulated in its current form. I have provided a couple of main areas where the work can be improved upon.

We thank the reviewer for their careful reading and feedback of our study, and we agree those are the main takeaway points of the study. We have now substantially changed the section detailing the atmospheric dynamics and circulation analogues to make the paper more logical and easier to follow. All references noted in our response here are listed at the end of our response document.

Major Considerations

1. The climate models used have well-known deficiencies in simulating multidecadal variability which likely needs to be emphasized for the pre-industrial CMIP5 comparisons. Previous studies have done work at establishing model credibility of low-frequency variability and these may be used to cull from the broader set of models.

We fully understand the concerns of the reviewer, in that multidecadal variability may be deficient in some models - we now make a note of this in the revised Methods, particularly in regards to Atlantic multidecadal variability. Note, however, that because our AMIP-type are forced by observed SSTs, they are not sensitive to the deficiencies in SST-related variability in the CMIP climate models. As with other studies (e.g., Tokinaga et al. 2017), we have decided to include the maximum number of models to capture as many heat wave events as possible. Decades with clustered heat extremes are obviously rare in the models (see next response), given there is no human controlled land-cover changes in the piControl simulations that would increase the risk of amplified heat extremes like those in the 1930s Dust Bowl or prolonged drought periods.

The main point of the CMIP5 part of the paper is to show that (1) heat extreme decades are produced in the (unforced) piControl simulations; and (2) these simulated decades typically occur during positive AMO and negative PDO phases, similar to what has been observed for present day heat waves (McKinnon et al. 2016).

a) Do models actually show that heat extremes are clustered decadal? Is this the basis for selecting results from GFDL-ESM2G?

Yes, most CMIP5 models produce heat extremes clustered into decades. Below, Figure R1 shows an example for GFDL-ESM2G over its entire piControl simulation (500 years). Each point indicates the number of heat extreme summers (black circle = heat wave duration; red circle = heat wave amplitude) in each decade that surpasses one standard deviation

above the long-term average. There are six separate decades where five summers (in that decade) feature either significant heat extremes in terms of duration or amplitude (or both). There is also one decade with six significant extreme summers. In fact, 14 of the 22 CMIP5 models have at least one decade with six significant summers, while three models produce one decade in their 500-year simulation with seven significant summers. Only one model, FGOALS-s2, does not produce at least one decade with five significant summers. We also note that the GFDL example is selected due to its surprising similarity to observations, see below.

Figure R1: Count of heat extreme metrics per decade, averaged over the central US (105°-85°W, 30°-50°N), that surpass one standard deviation above the long-term mean for the GFDL-ESM2G 500-yr piControl simulation. Shown are the count of events based on heat wave duration (black hollow circle) and heat wave amplitude (red solid circle). For example, a value of 5 equates to half of the decade in question featuring an extreme summer in terms of that particular heat wave metric. The points are based on a 10-year running sum, shifted ahead one year at the time.

We apologise for not properly explaining the significance of the GFDL-ESM2G model in the main body of our paper. We have now added “*The GFDL-ESM2G model is...an example of one such model decade which realistically matches the observations in terms of its SST patterns and heat wave activity (e.g., it breaks similar HWF records over a large spatial extent of the central US, as seen in the 1930s).*”

We have also rewritten the part of the Methods describing how we find the record-breaking heat wave decades. We show an example of this in Figure R2 below, highlighting the third century of piControl years for a selection of CMIP5 models (refer to the middle column of values in Table S1 of our manuscript). This figure presents the most common decades in that century with record-breaking heat wave frequency values over the central US. For observations, the 1930s is the most prominent decade (first two panels of Figure R2). GFDL-ESM2G is shown in the bottom row, with around 54% of the central US featuring record-breaking heat wave frequency values in its third piControl century. Other models

show record-breaking decades that are more sparsely located (e.g., bcc-bcsm1-1m). GFDL-ESM2G is the only model to break records across more than 50% of the northern and southern sections of the central US (as was observed), and why we highlight it in the paper. On a side note, it is worth noting that GFDL-ESM2G is one of the few CMIP5 models that produce an AMO-like multi-decadal signal that is similar to observations (Mann et al. 2020).

Figure R2: A selection of CMIP5 models showing the decade where each grid point recorded a record-breaking heat wave frequency for that century (model years 201-300). Observations from GHCN-D and BEST are shown in the first two top panels, highlighting the 1930s decade. Black dots cover the area of the most common decade of that century that experienced record-breaking extremes (e.g., GFDL-ESM2G shows stippling over most of central and northern part of the central US – this is decade centred on the year 91 (Year 291 out of 500)).

b) Lines 305-308 suggest that querying GCMs for HWF records clustered in a 10-year period is considered. Are there any key properties about GCMs that do this different than others and how they capture elements of decadal oceanic variability?

Thanks for raising these interesting points. One of the main properties in GCMs that will obviously control heat wave activity (both interannual and decadal) is the variability in the

Atlantic and Pacific, and their influence on drought, through teleconnections to rainfall. There is now a strong line of evidence that suggests multidecadal variability in Atlantic SSTs is one of the key factors in forcing multi-year climate variations, including droughts across the central US (Nigam et al. 2011; Sutton & Hodson 2005). Multi-decadal variability in the Pacific is also important (McCabe et al. 2004), which can be broken down into the contributions of tropical and extratropical Pacific SSTs (Schubert et al. 2004). It has been shown that CMIP5 models underestimate the variance and amplitudes of the Pacific and Atlantic multidecadal modes on time-scales greater than 20-years (Cheung et al. 2017, Mann et al. 2020; we now cite this study). This might be the reason why CMIP5 models struggle to simulate the correct precipitation over the central US in response to the AMO forcing in their historical simulations (Sheffield et al 2013). Yet, as previously stipulated, we use observed SSTs in our AMIP runs, so some of the effect of this is avoided.

Our analysis suggests that the CMIP5 models tend to simulate a warmer North Atlantic and cooler tropical Pacific in association with decades of elevated heat wave activity (see Figure R3 below). However, the response in the heat wave activity maximum is located too far in the south, whereas the observed response was further north. So like our AMIP runs and others (Cook et al. 2009; Schubert et al. 2004), models in general reproduce Dust Bowl-like events (although somewhat muted), even in the absence of external forcings like dust and land cover changes.

Figure R3: Summer SST anomaly and heat wave frequency patterns averaged across the most active heat wave summer decade per century over the central US for 22 CMIP5 piControl experiments.

2) The section on atmospheric circulation influences on spring and summer is challenging to follow and not very convincing.

We have noted this, and along with the other reviewer's suggestions, we have now made this easier to follow. There is now less of a focus on the month-to-month variations in the

circulation and more on the broad circulation during the heat extremes. We have also removed the section on the circulation analogues as we felt it was difficult for a reader to interpret and not very informative. We are further grateful for the suggestions below.

a) Fig S6 shows some evidence of below-normal precipitation during Apr-May for ATLh, although more egregious are the larger positive anomalies from the PACh which may be further emphasized in the results. I would suggest to not alter the magnitude of results (i.e., Observed patterns are reduced by a factor of 0.5). Is there are way to mask out signals that are insignificant (e.g., where the ensemble mean is not different from 0 w/confidence intervals?).

You're correct in your interpretation of the previous Fig. S6, that the PAC_{HIST} precipitation anomalies are strongly positive. As such, the precipitation and evaporative fraction differences ($\Delta[ATL_{HIST}, PAC_{HIST}]$) stem from the contributions of both ATL_{HIST} and PAC_{HIST}. We have decided to not show the previous Fig. S6, replacing it with daily and monthly climatology plots, for example, the absolute rainfall for ATL_{HIST} and PAC_{HIST} ensembles (see Figure R4 below), averaged over the southern-central US. This better shows the ATL_{HIST} ensemble precipitating less and the PAC_{HIST} ensemble precipitating more during the mid-spring to early summer months. This produces the significant precipitation difference seen in Apr-May (see the revised Fig. 4). There is less of a precipitation difference in the mid-late summer months when the hottest and longest heat waves emerge, meaning preconditioning is an important factor (e.g., as shown for observations; Donat et al. 2016).

Figure R4: Daily climatology of precipitation, averaged over 1930-1937 for the southern central US (105°-85°W, 30°-40°N) for a (red) five-member ATL_{HIST} ensemble, (blue) five-member PAC_{HIST} ensemble, and (black) 10-member HIST ensemble. The member range for the ATL_{HIST} and PAC_{HIST} ensembles, and monthly climatologies of GPCP (circles) and GHCN-D (asterisks) are also shown. Precipitation has been smoothed with a 15-day running mean.

b) Fig S7 and accompanying text (lines 140-155) are hard to follow and don't seem to be well supported by the data visualized. I see very weak spatial pattern correlation between ATLh and the Observed fields. Perhaps calculating this and reporting the values will help overcome what my eyes are drawn to. Notably, the persistent ridging across the West from Mar-June in observations collocated with anomalously high MSLP would deter the low-level influx of moisture from the Gulf of Mexico into the Great Plains which is key to spring-summer water vapor transport. It might help to quantify integrated water vapor fluxes with these simulations

(Fig S3e has some of this, but it is reported for region that doesn't totally correspond to the drought area. This could be slightly reworked.

We agree that this section was overly difficult to interpret and the old Figure S7 contained information that was potentially confusing to the reader. We have restructured this section considerably, with a focus on (1) the circulation during the hottest summer heat waves, instead of showing monthly spatial averages, and (2) the reasons why the ATL_{HIST} ensemble produces greater heat wave activity than the PAC_{HIST} ensemble. We now also show a spatial difference map of the 850 hPa moisture fluxes in the revised Fig. 4a (the Apr-May difference, $\Delta[\text{ATL}_{\text{HIST}}, \text{PAC}_{\text{HIST}}]$), as well as the absolute zonal and meridional components of the moisture fluxes in the ATL_{HIST} and PAC_{HIST} ensembles over the southern-central US: 105°-85°W, 30°-40°N (added to the Supplementary figures). This suggests that there are weaker northward moisture fluxes in the late spring in the ATL_{HIST} ensemble, compared to the PAC_{HIST} ensemble (Figure R5 below). This is consistent with a spring precipitation deficit shown in Figure R4 above and lower relative humidity (see Figure R11 in our response to Reviewer #3). The weaker northward moisture transport over southern Texas is consistent with the 850 hPa wind anomalies from 20CR reanalysis for the 1934 and 1936 March-June months, which preceded the hottest summers in the 1930s (Donat et al. 2016).

Figure R5: Monthly climatology of 850 hPa moisture flux, averaged over 1930-1937 and the southern central US (105°-85°W, 30°-40°N) for the (red) five-member ATL_{HIST} ensemble and (blue) five-member PAC_{HIST} ensemble. The member range for the ATL_{HIST} and PAC_{HIST} ensembles is also shown. The top profiles are the meridional component, while the bottom profiles are the zonal components. Positive values reflect northward and eastward moisture transport, respectively.

c) I understand the motivation for trying to parse out the contributions from each basin. But, it would be worthwhile considering an experiment that shows the full HIST simulation to potentially assess whether there is some interacting circulation pattern.

As we have restructured this section, we now show the Z500 and MSLP anomaly patterns for the week of the hottest heat wave for the two idealised ensembles (ATL_{HIST} and PAC_{HIST}) and the full HIST ensemble (see Figure R8 below in our response to Reviewer #2). We argue there is interaction taking place whereby the summer heat wave and spring precipitation responses in the HIST ensemble are controlled by the interplay between the

Atlantic and Pacific SST forcing. The Atlantic is forcing an elevated heat wave response (exacerbated through greater precipitation deficits), while the Pacific SSTs drive a dampened heat wave response (through positive precipitation anomalies). Our restructuring of the section now better outlines the circulation and precipitation differences between the ATL_{HIST} and PAC_{HIST} ensembles.

Minor considerations

1. Line 25, remove “pre-industrial”

This is now removed.

2. Line 37, “prolonged unusual synoptic conditions” is this well understood as being part of a blocking pattern through which land-surface interactions strengthen a large wave pattern?

Yes, here we’re referring to the blocking anticyclonic pattern, one that is unusual with respect to other drought periods (e.g., 1950s). We have now reworded this to: “...through land-atmosphere interactions contributing to summers dominated by a strong upper level ridge, allowing the extreme heat to build during the Dust Bowl drought”.

3. Line 82, Why chose 1930-1937?

These years were when the heat wave conditions were the most extreme, based on heat wave number, duration and amplitude (see Figure 4 from Cowan et al. 2017). Other studies have chosen 1932-38 (Hu et al. 2018), 1931-39 (Brönnimann et al. 2009), 1932-39 (Cook et al. 2011; Schubert et al. 2004) possibly because the prolonged drought conditions through to the autumn of 1939 and winter of 1939/40, however heat wave conditions were somewhat moderate after 1937. We now briefly explain our justification for our choice in the revised manuscript.

4. Tests to examine whether circulation only explain the magnitude of the heat extremes were a bit cumbersome to follow.

We apologise for this. We have decided to remove the section on the circulation analogues, as it does not fully represent the possible influence of the land-surface memory on surface temperature extremes, as the reviewer has noted below. We have now replaced it with a revised discussion on the circulation during the simulated heat wave events.

A few questions follow:

a) It is unclear whether analogs are selected using individual days or individual weeks. The latter seems more appropriate given the influence of land-surface memory on realized surface temperatures (e.g., short-term rapid drying with heat stress may further enhance Bowen ratio);

These were based on individual days, as the original idea here was to verify the daily temperature anomalies associated with the most similar daily circulation pattern. For the model analogue patterns these can be from any day within the summer season for the 1930s, but we found that it was quite common to find 3-4 day sequences of analogue patterns from the simulations. However, you are right in that it is unlikely the model (HIST simulation) would ever capture the observed temperature anomaly given the HIST analogues are made up of random set of patterns (i.e., days), some which would be part of a sequence of hot days and others that are not. Another issue that we discovered is that the same circulation pattern could be selected more than once, if it best matches the observed pattern for a sequence of days. For example, if Day #1 of an observed heat wave best matches a simulated

day (e.g., Day X in the model), then observed Day #2 and Day #3 (and so on) can also match simulated Day X. This means the ensemble average of the analogue patterns includes multiple non-unique days. This likely results in unintended biases in our analogue patterns. For that matter, and given the restructure listed in our previous responses, we have decided to remove the analogues part of the paper.

b) Are 500hPa heights or 500hPa height anomalies used? The prior might be better, but would likely bias results for models that have mean-state biases in Z500. The latter might correct for mean state anomalies, but have other problems.

As we now do not show the analogues part now, this is no longer an issue. Just out of interest, we used anomalies, as described in Jézéquel et al. 2018. Any mean-state biases in the model, which we know are a factor, would be present in all simulations (i.e., we're not comparing model against model).

c) I do not completely follow the justification for using the Mahalanobis distance. The Mahalanobis distance would be advantageous for multivariate analogs (e.g., joint consideration of MSLP and Z500).

As noted above, this is not an issue now, given we've removed the model analogues. Out of interest, originally we used root mean square error (RMSE), but were informed that Mahalanobis distance was more suited due to the different latitudinal and longitudinal covariance structures. This actually does not make a big difference. For example, using analogues calculated from the area-weighted RMSE minima produces similar Tmax anomalies to that from analogues calculated from the Mahalanobis minima (see Figure R6 below). Other studies focusing on heat waves on the daily-scale have used RMSE when focusing on either Z500 or MSLP (e.g., Jézéquel et al. 2018, Harrington et al. 2019).

Figure R6: As in old Figure 3a, but using analogues selected by the area-weighted root mean square error (RMSE) minima, instead of the Mahalanobis distance. It can be seen that the ATL_{HIST} Tmax anomalies are consistently higher than those in PAC_{HIST}.

5. Line 158, remove “mid-west”, also subsequent line is it fair to said anything here about cropland failures in coupled GCMs - GCMs are not explicitly simulating crop yields or failures, right?

Yes, you are correct. What we should be saying is that bare soil replacing C3 and C4 grass (i.e., vegetation) is reminiscent of crop failures (i.e., referring to the Cook et al. studies). Also, we have removed mid-west. Thanks for picking this up.

6. Line 162, Are these prescribed bare soil experiments dynamic such that soil can be barren in 1930 and recover through dynamic vegetation processes, or will bare soil in 1930 be bare in 1936?

The land fraction values are linearly interpolated between decades as in CMIP, instead of varying each year. The model does have dynamic vegetation, but the amount of the soil in each grid cell from 1930 to 1936 will linearly change depending on what the 1940 value is. This is common for coupled climate models. To highlight this, we now include a new Supplementary Figure, which shows the change in grass coverage and bare soil in our experiments (see Figure R9, right panel, in our response to Reviewer #3). We have also made our description of the bare soil simulation easier to understand, as this was not included in our previous manuscript. Thanks for bringing this to our attention.

7. Line 298, Repeat sentence.

Thanks for spotting this. Now deleted.

8. Line 184, Notably as well to changes in cropland practices and potentially the lack of recent extremes rivaling the 1930s is increased crop intensification and its corresponding role in increasing ET (Mueller et al., 2016). This might be thought of as sort of an opposing experiment to the bare soil approach used here.

We agree that this would be an interesting question but perhaps out of the scope of this study where there is limited space. Your suggestion is also in agreement with Alter et al. 2018 who showed that irrigation and intense agriculture is possibly a greater forcing than greenhouse gases and natural variability. We agree that this would make a nice experiment, particularly important for future projections of drought and their influence on cropping and potential yield losses (e.g., Glotter & Elliot, 2016). For reference, we have now added in the Mueller et al. 2016 reference towards the end of our discussion section.

9. Fig 3b, The units of Mahalanobis distance are not useful. Consider reporting these as standardized units instead.

Thanks for noting this. Given we do not show the analogues in the revised manuscript, this is no longer an issue. That said, we certainly agree that magnitudes of 10^7 were not overly meaningful.

Reviewer #2 (Remarks to the Author):

This paper discuss the relative contribution SST forcing, atmospheric circulation and land cover changes on the 1930 dust bowl. The study is interesting, evidence are convincing, the sensitivity experiment set-up is complete. I think it is an interesting study. Despite of this, I don't think this study fits in Nature communication, especially for a question of format. My feeling is that the text is too dense and somehow hard to follow, I think this large and complete study would fit much better in a longer format, such as Journal of Climate. If the author prefers to publish in Nature Communication, I would suggest them to think a bit more carefully which figure and text should be part of supplementary material.

We thank the reviewer for their detailed and thorough assessment. We appreciate their views on the study's fit in Nature Communications. We strongly believe the results are of broad enough interest to publish in Nature Communications, as very little has been published on the Dust Bowl heat waves, combining coupled model results with idealised and sensitivity experiments. To reduce the size of the paper, we have removed the section on the circulation analogues and made the circulation and spring conditions section easier to follow. We have also cut the number of Supplementary figure panels and tables and restructured the manuscript accordingly. This should make the paper easier to interpret.

Below my detailed comments:

Scientific:

The model biases and set-up limitation are not discuss comprehensively in the text, some biases are mentioned, but I miss a paragraph clearly explaining the biases of the HadGEM3: wet spring bias, misrepresentation of the heat low, land atmosphere coupling, warm bias, too weak temperature anomalies, and a clear discussion of how it could affects the results of the present study.

We acknowledge that we only briefly covered the model limitations and biases in the manuscript. We now discuss these biases and their implications more broadly in the revised manuscript. Also, we discuss caveats associated with the HadGEM3 experimental set-up, particularly with respect to the bare soil sensitivity experiments (these are placed in the Methods section), noting that they are not meant to accurately represent the observed conditions, but more to gain an understanding of what happens to heat wave activity in our model simulations if the C3 and C4 grasses are replaced with bare soil.

Obviously the main issue is the warm mean state bias over the central US in the summer, particularly over the southern central US (30°-40°N; see Figure R7 below) and potentially the wet bias too. This appears to be a common issue amongst the set of AGCMs that were analysed in this study (again refer to Figure R7 below). Recent research further suggests that this continues to plague CMIP5-generation models, such that numerous models over-amplify hot extremes, even in wet regions like the southern US (Ukkola et al. 2018). As mean-state and variability biases are inherent in nearly all models, and to discard a model that is quite competitive against other AGCMs would open up similar studies using AMIP experiments to the same criticism. We further argue that the wet biases do not significantly influence our results, given we are making comparisons within the same model (i.e. Atlantic versus Pacific) and its own climatology. In terms of trying to understand the

mechanism for the heat waves (e.g., dry springs), our main conclusions are still meaningful and give some important insight into the role of SST anomalies. We certainly acknowledge that there are a range of responses across different models, noting that, as Cook et al. 2011 state "*most models are only able to partially reproduce the pattern, intensity, and location of the Dust Bowl precipitation anomalies*".

Figure R7: Time series of (top row) mean summer (Jun-Aug) surface temperature, averaged over the southern Great Plains (105°-85°W, 30°-40°N), and (bottom row) mean spring (Mar-May) precipitation, averaged over the same region, for 1916-1955. The long-term average is shown as the horizontal line separating the two colours. The numbers in brackets refer to the number of ensemble members for each AGCM. All models are too wet in spring (observed value during Dust Bowl is 2.2 mm/day), and too warm in summer (observed value during Dust Bowl is 297 K).

I would suggest the author to include Ardilouze et al. (2017), which have discussed the relation between soil moisture and heat wave in the US great plain and how biases in soil-atmosphere coupling lead to misrepresentation of heat wave.

Thanks for the reference. This is now been added in with our discussion on the biases in land-surface models.

Role of land cover changes: The set-up described here is pretty simple and drastic, how realistic is this set-up? The limitation of the sensitivity experiments should be more carefully discussed. **As with our first response, we now discuss this point more clearly in our revised manuscript. The set-up is supposed to be somewhat rudimentary, given how little information there is about actual bare soil amounts in the 1930s. As remarked by Cook et al. 2009, “there are few spatial estimates of crop failure during the period (1930s), although evidence suggests quite high and significant abandonment of agricultural lands and loss of vegetation cover during the period”. They refer to a study by Hansen et al. 2007, who show a basic map of wind erosion across a smaller Great Plains region. Figure 2c of Cook et al. 2009, shows the devegetation fraction from crop removal that they used to force their AGCM experiments. They use similar devegetation fractions (10-20%) to what we use in our 50% soil run, although they assume “no loss of natural vegetation during the drought”. We made the assumption that both C3 and C4 grasses would reduce by similar amounts to account for the increase in bare soil. We also feel our estimates are not too dissimilar to those shown in Figure 5 of Gutmann et al. 2016 for the year 1934 - they get their estimates**

from an agricultural census. Our devegetation estimates are shown in Figure R9 (in our response to Reviewer #3). We now have decided to show these devegetation estimates in our revised manuscript's Supplementary section, as well as how the changes are linearly interpolated across the decade from 1930 to 1940 (see Figure R9, right panel). We also note in our revised paper, that all estimates of the bare soil return to 15% by 1940, meaning the 30%, 50% and 80% soil runs refers to the approximate percentage of the central US covered by that surface type in 1930 (which reduces as the decade progresses). We apologise for not making this clear in our original submission.

L180-181: From the text it is not clear to me what make you conclude the land cover change is the most important contributor, further sensitivity experiments, with change imposed SST and change in land cover would be needed to determine the respective contribution of each.

Perhaps we were overly enthusiastic in our view that land cover changes were the most important contribution, given the uncertainties around the spatial extent and magnitude of the forcing. What we mean is that in our model simulations, the greatest change in heat wave activity stems from changing the bare soil amount. We see this, for example, in the measure of heat wave duration, where changing the bare soil amount from 30% to 50% over the central US, results in an increase in the longest heat waves by 10 days (approx. 5 days to 15 days). Similar spikes are seen for heat wave frequency. We agree that we would have to conduct more realistic sensitivity experiments to quantify the importance of SSTs versus land cover changes. Unfortunately, estimates of land cover in the lead-up to (and during) the 1930s is hugely uncertain (e.g., Hu et al. 2018), and biases discussed above will make it challenging to provide relative contributions of SST forcing and devegetation to the observed events - yet the simulations indicate a potential for very large enhancement of heat waves due to vegetation responses.

L187-190: I don't understand which result support this conclusion.

Thanks for spotting this error. This sentence remained from an earlier redundant version of the paper. We have since removed this sentence.

Structure:

Role of spring and summer circulation: This paragraph is extremely dense, here I clearly miss in the main text, the visualization of the analog pattern of ATL_{HIST} and PAC_{HIST}.

We agree that this section is difficult to follow, as picked up by Reviewer #1. For that reason, we have undertaken a major restructure of the section, beginning with removing the circulation analogues analysis. We have replaced this with a figure showing the circulation anomalies (Z500 and MSLP) associated with the hottest heat waves in the reanalysis and HadGEM3 ensembles (see Figure R8 below). We argue this is easier to understand, and takes into account the day-to-day memory in the land surface (at least over a week-long period) which could potentially feed back through to the circulation. We also now show the daily climatologies of Z500 and MSLP, averaged over 1930-1937 and the eastern US (105°-75°W, 30°-50°N) in a revised Supplementary Figure S4. This also highlights the difference in the Z500 between the ATL_{HIST} and PAC_{HIST} ensembles, clearly showing the stronger mid-tropospheric ridging in the spring-time months in ATL_{HIST} ensemble (compared to the PAC_{HIST} ensemble).

Figure S4, could be replaced by the ensemble mean of Z500 analog pattern for observations, and the 2 sensitivity experiments, and I think this figure should be part of the main text to help the reader. In addition, figure 3b and c shows significant differences in the Mahalanobis distance for 2 years during summer, which is not commented in the text.

Thanks for this great suggestion. We have now decided to replace our old Figure 3 with Figure R8 below, which shows the ensemble mean Z500 and MSLP anomaly patterns for the 7-day period of the hottest heat wave (for each experiment, and for each summer over 1930-1937). The old Figure S4 has been removed. As we no longer focus on the analogues, we have removed the points referring to the Mahalanobis distance. Thanks for initiating this change in our manuscript, which greatly improves the structure and flow of our study.

Figure R8: MSLP (contour) and Z500 (colour) anomalies, averaged over a 7-day period from the start of the hottest summer heat over the central US, and averaged over 1930-1937. Shown are (a) 20CR, (b) ten-member HIST ensemble, (c) five-member ATL_{HIST} ensemble and (d) five-member PAC_{HIST} ensemble. The thick line indicates the 0 hPa MSLP anomaly, while the solid (dashed) lines indicate positive (negative) MSLP anomalies.

From L140 to 156: the text is really hard to follow. My feeling is that model biases and misrepresentation of heat low and precipitation in HIST should be discussed separately from the results of sensitivity experiments.

We acknowledge that this part was not well worded. As with our previous response to your suggestion, we now discuss the model biases in greater detail, in a separate paragraph.

Minor:

L 140: I understand that “spring” is missing here.

You’re correct, although we’ve decided to rewrite this whole section.

L156-157: Missing transition

We have added in a transition sentence.

L86: could this be related to the bias described in Ardilouze et al. 2017

Yes, this could definitely be the case given it's the same Great Plains region. We have now added this in while referencing the Ardilouze et al. study.

Reviewer #3 (Remarks to the Author):

Review of Ocean and land forcing of the record-breaking Dust Bowl heat waves across central United States.

This is an interesting and important topic especially given the critical relevance of drought, heatwaves, and our need to understand the interactions and cascading impacts between such processes. Overall the manuscript is well written and well communicated. However, there are a number of questions and clarifications the authors should address prior to formal publication.

We are appreciative of the reviewers comments and thought-provoking suggestions. We feel now that we provide a more sound interpretation of our results. We also now provide more information on our bare soil sensitivity runs, including spatial maps of the devegetation levels and the how the model's land surface types change over the 1930s.

The choice of using the 1916-1955 model climatology is not specifically justified in the manuscript. Further, the use of this window is potentially problematic given a large portion of the study domain was locked in another “decadal” drought during the 1950s. In fact, in portions of the Southern Great Plains (e.g., Oklahoma, Texas, etc.) the overall magnitude of the thermal anomaly(ies) during the JJA period in the 1950s was as large or greater than during the dust bowl period. This may be (in part) why this region of the domain does not show as strong of a heat wave signature during the dustbowl interval. Can the authors provide a more detailed justification for using this temporal window versus another?

Thanks for bringing this point up. We acknowledge the choice of this 40-year climatology will include a portion of the 1950s drought in the Southern Great Plains. We originally performed these simulations with the intention of focusing on the period prior to the 1950s (i.e., termed the Anthropocene). As such, we made a decision to terminate the simulations before a prominent greenhouse gas-forced signal emerged. Hence, these simulations run through to the mid-1950s, and we use all available model years to give us the longest climatology possible. In Figure 2, in the comparison of the observations to the HadGEM3 model results, both are referenced to the same period, so we argue this is taking a more conservative approach (i.e., we're not cherry-picking periods to artificially boost our anomalies). We now provide more details on our choice of climatology in the paper.

In the experiments focused on devegetation, and considering the domain used, large portions contain forested areas versus grassland. How were these areas treated? Why do areas that are heavily forested (e.g., the southeast portion of the domain) see such a dramatic HWF signature during the devegetation experiments versus the grassland ecosystems further to the west (nearly double the total number of “days”)?

Those are both interesting questions. Just to be clear, the JULES land-surface scheme has each land point subdivided into five vegetation types: broadleaf trees, needle-leaved trees, temperate C3 grass, tropical C4 grass and shrubs. There are also four non-vegetated surface types: urban, inland lakes/rivers, bare soil and land ice. What we did was convert only the C3 and C4 grasses into bare soil at different fractions. Figure R9 below shows the amount of bare soil percentage values (i.e., at the expense of grass + HIST baseline), when the bare soil amount was increased to 30%, 50% and 80% (at the expense of C3 and C4 grasses) across the central US. In our original manuscript, we incorrectly stated the devegetation values as 17%, 33% and 66%. As we have revised in our manuscript, this should be approximately 25%, 50% and 100% averaged across the central US. That refers

to the percentage loss of grass, and not trees, so there is still vegetation. This still means other surface types like trees remain the same in each grid cell. Our intention was to perform these rather rudimentary sensitivity experiments, similar to those performed by Cook et al. 2009. As we have noted in our study, it's likely that HadGEM3's overestimation of the strength of its land-atmosphere coupling (i.e., warm bias) causes over-amplified summer heat extremes. This is very much the case in the bare soil runs, when comparing the heat wave metrics (e.g., average temperatures exceeding 55°C). We must also remember that, given there is only one sensitivity experiment per soil fraction, it's possible that internal model variability plays a role, although the changes simulated are substantially larger than internal variations between HIST simulations.

Figure R9: Bare soil and grass fractions in bare soil simulations. (left panels) Percentage of bare soil per grid cell over the central US in 1930, in the bare soil simulations where the regional average is (top left) 30%, (middle left) 50%, and (bottom left) 80%. The percentage of bare soil is calculated as the amount of soil in the baseline HIST experiment (see right panel) plus the amount from the conversion of the combined C3 and C4 grass fractions to bare soil. Land-type fractions have a decadal temporal resolution in the model and are linearly interpolated across decades. The right hand-side vertical axis is land-surface

type fraction, so is unitless.

As to why the southeast of the domain simulates the largest HWF values (compared to the areas where grass removal occurs), this is likely due to two reasons:

- (1) the nature of heat waves in response to warmer temperatures in general. We know that from future warming projections, tropical regions experience a greater increase in HWF, due to their low interannual variability (i.e., there is less diurnal and seasonal variation in temperature than in the midlatitudes). Further south in the tropics, for a small increase in temperatures due to an increase in bare soil, there is greater potential for a larger response in the number of heat wave days.
- (2) biases in the model. As can be seen from the broad HWF response in HadGEM3, and in other AGCMs (and for GFDL-ESM2G in its piControl simulation), the region of the maximum signal is placed in the southern part of the central US. It is therefore likely that model biases, accentuated by bare soil, trigger a greater heat wave response in the south (centered around 97°W-84°W). We see this more clearly when looking at the HWF in the 80% soil run, which is not included in our study but shown below in Figure R10. Here we see greater HWF values further south, but not necessarily isolated to the southeast.

We now briefly cover this point in the section discussing the biases in the revised paper.

Figure R10: Average summer HWF over the central US for 1930-1937 for the 80% soil run in HadGEM3.

Perhaps this is just semantics, but drought is never formally defined in the manuscript. Additionally, instead of using “drought” in the Spring precipitation analyses, the authors note “precipitation deficits”. So, this leads to the question: Did drought occur during the Spring precipitation period as implied?

That is definitely a valid point. We have now added in a brief half sentence, early in the Introduction, with a reference to our earlier study (Cowan et al. 2017), where we defined the drought using both the SPI and PDSI, and showed that both summers and springs in the 1930s were anomalously dry (e.g., 1930, 1931, 1934, 1936). On the second part to your question, here our focus was less on what constitutes a drought, and more on the role of spring-time precipitation deficits in pre-conditioning the summer heat extremes. We could

call it drought, although we don't really want the paper's focus to be on does (or does not) constitute a drought. We also know from the work of Donat et al. 2016 that "*unprecedented summer heat during the Dust Bowl years was likely exacerbated by land-surface feedbacks associated with springtime precipitation deficits*". In their study, they also focus on a springtime deficit for March-June. Furthermore, studies like Schubert et al. 2004 clearly show a spring-time deficit in the order of -0.15 mm/day, averaged over 1932-1938. What we clearly show now in our revised manuscript, is the HadGEM3 simulations forced only by varying Atlantic SSTs, produce a greater late spring rainfall deficit than the simulations forced by Pacific SSTs (see Figure R4 in our response to Reviewer #1). This in turn leads to smaller evaporative fractions which carry through to summer, and hence prime the central US for enhanced heat wave activity. This result is consistent with what is shown for observations for 1934 and 1936 in Donat et al. 2016.

Additionally, timing (as noted by the authors) of precipitation is critical in the study domain because of the asynchronous nature of precipitation/temperature in the region (see Flanagan et al. 2017). Thus, even "normal" or even above normal precipitation occurring earlier in the year could have a similar net effect in the JJA period. In other words, timing and magnitude of precipitation matter.

Absolutely correct. Thanks for bringing this to our attention. We now show the daily precipitation from the simulations (see Figure R4 in our response to Reviewer #1), which clearly shows the rainfall deficits occurring from mid-April through to the start of June. This would not be captured focusing purely on seasonal or half-year averages.

Revisiting the vegetation portion of the study, the authors approach the link between heatwaves and terrestrial feedbacks focused on the Dust Bowl by removing the vegetation – for obvious reasons. An alternative experiment would be to simply desiccate the existing vegetation versus replacing with bare soil. Are the results similar? The reason for this approach is that, especially when considering future potential outcomes, the lessons learned from the 1930s in terms of land management have yielded starkly different practices and it is highly unlikely that such bare soil type of conditions would ever re-occur in the central US in the future. However, extreme Spring drought could lead to large-scale vegetation desiccation regardless of human efforts. In this scenario, would the results be similar?

Again, another really interesting question. Essentially what this would mean is a complete breakdown of the evaporation-transpiration cycle in the vegetation, as you would be preventing grasses (i.e., just shallow-rooted vegetation) from taking up moisture. Unfortunately, for our study, it is out of scope of one of our main questions, that is, the role of bare soil on the Dust Bowl heat waves. We feel that this would be better answered in a separate study where a more comprehensive experimental framework could be developed and tested. One could potentially replace tropical grasses with temperate grasses in tropical regions to see the impact this would have on moisture uptake, or replace deeper rooted plants with shallow-rooted plants.

In our study, two of the sensitivity experiments (30% and 50% soil) still have a sizeable fraction of grass cover (approximately 50% and 33% in 1930 that increases to 66% by 1940), so it is not unforeseeable that conditions like this could re-occur given the current rates of ground-water depletion in the southern High Plains. There is evidence to suggest that more than a third of this region may not be able to support irrigation within 30-years (Scanlon et al. 2017). Another study has pointed to projected mid-century temperature rises (i.e., with average rainfall) that alone could lead to crop losses similar to those during the Dust Bowl drought (Glotter & Elliot 2016). So even if "Dust Bowl-like" conditions rarely recur, it is highly likely that heat waves of the intensity observed in the 1930s will eventuate

and be surpassed, given future global temperature projections from the latest CMIP6 exceeding 4°C by 2100 (Figure 8 of Grose et al. 2020).

We reiterate that our experiments have been designed to be as simple as possible and focus on the broad implications of these heat waves under extreme conditions of land-use. The experiments provide us with only a rough estimate of the sensitivity of heat wave conditions in *our* model to increasing fractions of bare soil. A finer-scale assessment using Weather Research and Forecasting (WRF) has been undertaken by Hu et al. 2018, where they found land-cover changes in the 1930s makes “*the Great Plains more susceptible to drought*”. They found that replacing pre-settlement native pastures with dryland croplands and pastures in the 1930s caused a precipitation deficits across the Great Plains. As they state “*A direct consequence of these changes was reduced soil water storage ... and enhanced surface evapotranspiration in crops.*” Thanks to your suggestion, we have now added in a brief comment referring to the Hu et al. 2018 results (using a different set-up) versus our set-up of increasing bare soil amounts.

This study uses a heatwave definition solely based on temperature. Yet, there is increased awareness that the combination of extreme temperature combined with anomalous atmospheric humidity has greater human impacts and are likely more concern into the future. While it may make little difference in the overall results in this case, would a more advanced HWF metric incorporating humidity (apparent temperature) be more valid?

We agree that this would be a more valid index if the focus was purely on human morbidity and mortality, as we know that the combined effects of temperature and humidity are the biggest factors in heat waves, particularly in tropical regions. Just out of interest, below (Figure R11), we have plotted out the daily climatology of relative humidity, averaged over the northern and southern central US and 1930-1937, for the ATL_{HIST} and PAC_{HIST} ensembles. Relative humidity is a major factor in the apparent temperature definition. What this analysis shows is that there is very little difference in relative humidity between the two ensembles in the northern part of the central US, however the ATL_{HIST} ensemble is much drier than the PAC_{HIST} ensemble, particularly in late spring and summer months. This is consistent with the precipitation climatology for 1930-1937, now shown in the revised manuscript as a replacement for the previous Figure S3. We also now make a brief comment on the relative humidity in the revised version.

Relative humidity at 1.5 m

Figure R11: Daily climatology (smoothed with a 15-day running mean) of relative humidity, averaged over the (top) northern-central US (105°-85°W, 40°-50°N), and (bottom) southern-central US (105°-85°W, 30°-40°N) and over 1930-1937, for a five-member ATL_{HIST} ensemble (red) and five-member PAC_{HIST} ensemble (blue). The figure shows that there is a greater relative humidity difference between the ensemble means in the south, but not the north.

The study also generally skirts the land-atmosphere impacts of the analysis while more robustly discussing the SST/teleconnections. This is likely due to uncertainties in the models (as discussed by the authors) and the discussion is primarily focused on one-way processes whereby dry surfaces yield greater HWF. Yet there are known, strong linkages between the surface and atmosphere in the study domain (e.g., Koster et al. 2004, Guo and Dirmeyer 2013, etc.) and the linkages are stronger during drought events (e.g., Basara and Christian 2018, Wakefield et al. 2019). Further, there is likely a pathway to the feedback that noted in Miralles et al. 2018. Do the results of this study show the full pathways between such features including local humidity and vertical profiles that would inhibit precipitation?

Firstly, thank you for the list of relevant studies, many of which we were unaware of. This is an important topic of discussion, and one that could easily lend itself to a separate study. What we judge from our summer circulation results is that the drier spring conditions do not necessarily create warmer summer conditions and more heat wave days through a land-atmosphere feedback. In our original manuscript, we showed there was very little distinguishing the summer mid-tropospheric circulation between the ATL_{HIST} and PAC_{HIST} ensembles, yet the ATL_{HIST} ensemble consistently produces warmer temperatures. In our revised Figure 3 (see Figure R8 above), the circulation differences between the ATL_{HIST} and PAC_{HIST} ensembles for the week of the hottest heat waves shows stronger mid-tropospheric ridging across the much of central and eastern US (see Figure R12 below). This is not seen,

however, across the full summer season, consistent with similarities in summer-time precipitation and moisture fluxes in the SST-forced ensembles.

Figure R12: Difference in Z500 anomalies between the ATL_{HIST} and PAC_{HIST} ensembles (Figure R7c,d), averaged over a 7-day period from the start of the hottest summer heat over the central US, and averaged over 1930-1937. Units are in metres.

One would expect that land degradation and increased dust (i.e., our bare soil simulations) would lead to a reduction in the net surface radiation as a result of increased surface albedo, leading to increased subsidence and the inhibition of local convection. This would drive a reduction in precipitation and an increase in surface temperatures, further drying out the soil. Interestingly, in response to increasing bare soil, and subsequently, increased dust, we do not see a systematic reduction in precipitation (see our revised Figure 4) or low level divergence in the case of positive Z500 anomalies. We do see a deepening surface low, which is associated with the hotter temperatures. Below, in Figure R13, we also see that, over the southern central US during the 1930s, evaporation from soil decreases with increasing bare soil (consistent with the crop removal experiments performed by Cook et al. 2009), and there is a deepening the atmospheric boundary layer thickness increases, as expected with warmer temperatures and drier soils (e.g., Alexander 2011). The surface air is also drier in summer with lower relative humidities (Figure R13e). This carries through the summer months into autumn.

1930-1937 climatology

Figure R13: Annual cycle of Dust Bowl surface and atmospheric conditions in bare soil experiments, including (a) evaporation from soil, (b) atmospheric boundary layer thickness, (c) surface albedo, (d) convective rainfall, and (e) daily surface relative humidity. All variables are averaged over 1930-1937 for the southern-central US (105°-85°W, 30°-40°N) for the 30% (orange), 50% (red) and 80% (pink) soil runs. Relative humidity is smoothed using a 15-day running average.

However, we do not see any difference in the surface albedo (Figure R13c), or reduction in convective rainfall from the 50% to 80% soil run, consistent with a lack of a systematic reduction in moisture fluxes and mid-tropospheric ridging (referring to revised Suppl. Figure S7). This tells us that by evoking bare soils, we are only seeing an effect on the local-scale conditions, where drying soils in late spring and summer feed into an increase in temperatures and heat wave activity through the summer months. This appears to have little effect on the large-scale atmospheric patterns beyond the boundary layer.

The above results appear consistent with the experimental results from Cook et al. 2009 in their SST + CROP devegetation simulations, where they found virtually no change in precipitation over the Great Plains, but increases in surface temperature. However, they did find an upper-level convergence and low-level divergence response in the atmosphere, and subsequently a large-scale precipitation response over the Great Plains in their SST+DUST simulations. We now reflect on this in our revised manuscript, noting that our results appear consistent with similar devegetation experiments, but at odds with imposed dust experiments. This points to likely model deficiencies in HadGEM3.

Have the authors explored additional analogs to demonstrate the consistency of the results? One scenario could include the decadal drought of the 1950s previously mentioned. This is especially interesting in that many of the lessons learned from the Dust Bowl were employed before and during the drought with overall less severe terrestrial impacts even during pronounced and excessive heat. Additionally, more recently while decadal drought has not occurred in the region, significant annual-type droughts have including 2011 in the southern part of the domain (consistent with the 1950s drought and extreme heat) and 2012 (similar to the 1930s and also included heatwaves).

We originally focused on conditions outside of the 1930s, however previous reviews stated this somewhat detracts from our focus on the Dust Bowl period. What we found in this analysis of the period outside of the 1930s was that Pacific SSTs had a more dominant role in heat waves (i.e., mostly in the 1950s), which is consistent with prior research (e.g., Cook et al. 2011). It is worth noting that other studies have looked at paleo-records to find Dust Bowl drought analogs (unfortunately not heat waves), but as stated in Cook et al. 2011: similar droughts can “be identified; they are quite rare, however, and none fully capture the combined magnitude, duration, and geographical extent of the Dust Bowl”.

Finally, the study domain in the central US is actually quite large and incorporates approximately six regional climate regions defined by Karl and Koss (1984). Would the results hold if the domain size was changed or divided into the different subregions noted by Karl and Koss?

This region was chosen as it encompasses the region where the main extreme heat maxima, in terms of HWF, are located. This was also the region chosen by Donat et al. 2016 with the highest frequency of hot days during the July and August in the 1930s. We realise it is quite a large region, and our HadGEM3 simulations tend to place the heat wave maxima more towards the south. Hence it makes sense to focus on a reduced area. That being the case, we now show the differences in precipitation, evaporative fraction, and moisture fluxes over the smaller southern portion of the region (30°-40°N) in our HadGEM3 simulations. We agree that showing our results over a wider region doesn't bring out the strongest differences between the ATL_{HIST} and PAC_{HIST} ensembles.

Additional comments:

Why does the western part of Lake Superior experience a dramatic increase in HWF during the devegetation experiments?

This is just a masking error. We now have fixed this. Thanks for spotting the error.

Line 233 – an end bracket is needed.

Now added. Thanks.

Line 269 – need a “the” between “in” and “observations”.

Added.

References:

Basara, J. B., and J. I. Christian, 2018: Seasonal and interannual variability of land–atmosphere coupling across the Southern Great Plains of North America using the North American regional reanalysis. *International Journal of Climatology*, 38, 964–978.

Guo Z, Dirmeyer PA. 2013. Interannual variability of land–atmosphere coupling strength. *J. Hydrometeorol.* 14(5): 1636– 1646.

Flanagan, P. X., J. B. Basara, and X. Xiao, 2017: Long-term analysis of the asynchronicity between temperature and precipitation maxima in the United States Great Plains. *International Journal of Climatology*, 37, 3919-3933.

Karl T R and Koss W J 1984 Regional and National Monthly, Seasonal, and Annual Temperature Weighted by Area, 1895-1983 (Historical Climatology Series 4–3) (National Data Center: Asheville, NC) p 38

Koster RD, Dirmeyer PA, Guo Z, Bonan G, Chan E, Cox P, Gordon CT, Kanae S, Kowalczyk E, Lawrence D, Liu P, Lu C□H, Malyshev S, McAvaney B, Mitchell K, Mocko D, Oki T, Oleson KW, Pitman A, Sud YC, Taylor CM, Verseghy D, Vasic R, Xue Y, Yamada T. 2004. Regions of strong coupling between soil moisture and precipitation. *Science* 305(5687): 1138– 1140.

Miralles DG, Gentile P, Seneviratne SI, Teuling AJ. Land-atmospheric feedbacks during droughts and heatwaves: state of the science and current challenges. *Ann N Y Acad Sci.* 2019;1436(1):19–35. doi:10.1111/nyas.13912

Wakefield, R.A., J.B. Basara, J.C. Furtado, B.G. Illston, C.R. Ferguson, and P.M. Klein, 2019: A Modified Framework for Quantifying Land-Atmosphere Covariability during Hydrometeorological and Soil Wetness Extremes in Oklahoma. *J. Appl. Meteor. Climatol.*, 58, 1465–1483.

References in our response:

Alexander, L. (2011): Extreme heat rooted in dry soils. *Nature Geosci* 4, 12–13.
<https://doi.org/10.1038/ngeo1045>

Alter, R. E., Douglas, H. C., Winter, J. M. & Eltahir, E. A. B. (2018): Twentieth Century Regional Climate Change During the Summer in the Central United States Attributed to Agricultural Intensification. *Geophys. Res. Lett.* 45, 1586–1594.

Brönnimann S, Sticker A, Griesner T, Ewen T, Grant AN, Fischer AM, Schraner M, Peter T, Rozanov E, Ross T (2009) Exceptional atmospheric circulation during the “Dust Bowl”. *Geophys Res Lett* 36:L08802

Cheung, A.H., M.E. Mann, B.A. Steinman, L.M. Frankcombe, M.H. England, and S.K. Miller, (2017): Comparison of Low-Frequency Internal Climate Variability in CMIP5 Models and Observations. *J. Climate*, 30, 4763–4776.

Cook, B. I., Miller, R. L. & Seager, R. (2009): Amplification of the North American ‘Dust Bowl’ drought through human-induced land degradation. *Proc. Natl. Acad. Sci. U. S. A.* 106, 4997–5001.

Cook BI, Seager R, Miller RL (2011) Atmospheric circulation anomalies during two persistent North American droughts: 1932–1939 and 1948–1957. *Clim Dyn* 36:2339–2355

Cowan, T. et al. (2017): Factors Contributing to Record-Breaking Heat Waves over the Great Plains during the 1930s Dust Bowl. *J. Clim.* 30, 2437–2461.

Donat, M. G. et al. (2016): Extraordinary heat during the 1930s US Dust Bowl and associated large-scale conditions. *Clim. Dyn.* 46, 413–426.

Glotter, M. & Elliott, J. (2016): Simulating US agriculture in a modern Dust Bowl drought. *Nat. Plants* 16193, 1–6.

- Grose, M. et al. (2020): Insights from CMIP6 for Australia's future climate. *Earth's Future*, <https://doi.org/10.1002/essoar.10501525.1>.
- Gutmann, M. P. et al. (2016): Migration in the 1930s : Beyond the Dust Bowl. *Soc. Sci. Hist.* 707–740 (2016). doi:10.1017/ssh.2016.28
- Hansen ZK, Libecap GD (2004): Small farms, externalities, and the Dust Bowl of the 1930s. *J Polit Econ* 112:665–694.
- Harrington LJ, Otto FEL, Cowan T, and Hegerl, GC (2019): Circulation analogues and uncertainty in the time-evolution of extreme event probabilities: evidence from the 1947 Central European heatwave *Clim. Dyn.* 53 2229.
- Hu, Q., Torres-Alavez, J. A. & Van Den Broeke, M. S. (2018): Land-cover change and the ‘Dust Bowl’ drought in the U.S. Great Plains. *J. Clim.* 31, 4657–4667.
- Jézéquel A, Yiou P and Radanovics S (2018): Role of circulation in European heatwaves using flow analogues, *Clim. Dyn.* 50 1145–59
- Mann, M.E., Steinman, B.A. & Miller, S.K. (2020): Absence of internal multidecadal and interdecadal oscillations in climate model simulations. *Nature Commun.* 11, 49, doi:10.1038/s41467-019-13823-w
- McCabe, G. J., Palecki, M. A. & Betancourt, J. L. (2004): Pacific and Atlantic Ocean influences on multidecadal drought frequency in the United States. *Proc. Natl. Acad. Sci.* 101, 4136–4141.
- McKinnon, K. A., Rhines, A., Tingley, M. P. & Huybers, P. (2016): Long-lead predictions of eastern United States hot days from Pacific sea surface temperatures. *Nat. Geosci.* 9, 389–396.
- Nigam, S., B. Guan, and A. Ruiz-Barradas, (2011): Key role of the Atlantic Multidecadal Oscillation in 20th century drought and wet periods over the Great Plains. *Geophys. Res. Lett.*, 38, doi:10.1029/2011GL048650.
- Scanlon, B. R. et al. (2012): Groundwater depletion and sustainability of irrigation in the US High Plains and Central Valley. *Proc. Natl. Acad. Sci.* 109, 9320–9325.
- Schubert SD, Suarez MJ, Pegion PJ, Koster RD, Bacmeister JT (2004) On the cause of the 1930s Dust Bowl. *Science* 303:1855–1859.
- Sheffield, J. et al., (2013): North American Climate in CMIP5 Experiments. Part II: Evaluation of Historical Simulations of Intraseasonal to Decadal Variability. *J. Climate*, 26, 9247–9290.
- Sutton RT & Hodson DL (2005) Atlantic Ocean forcing of North American and European summer climate. *Science* 309:115–118
- Tokinaga, H., Xie, S.-P. & Mukougawa, H. (2017): Early 20th-century Arctic warming intensified by Pacific and Atlantic multidecadal variability. *Proc. Natl. Acad. Sci.* 114, 6227–6232.

Reviewers' comments second round:

Reviewer #1 (Remarks to the Author):

The authors have provided a thorough and thoughtful revision and addressed most of my outstanding questions. I am generally satisfied with this analysis but highlight below a few potential areas to strengthen the results using mainly statistical analyses and integration of results from the literature. While the authors did look at some attributes from multiple GCMs, their main conclusions seem to come from a single GCM ; this is not necessarily bad, but might benefit from a few additional tests for robustness and, where possible, comparisons to other model simulations. For example, it is unclear whether Figure 1c shows the result from one ensemble member or whether the postulation that GFDL-ESM2G captures decadal peaks in heatwaves is supported using larger ensemble approach. I do appreciate the additional figures provided in the response to reviewers, but it made me further question if the postulation that such peaks in heat waves are really decadal in nature in the GCMs. There could be a short statistical analysis that queries whether the decadal peaks in heat waves as simulated by the models happen at rates more significantly more frequently than if heat events were simulated through some random process without serial correlation. Similarly, Figure R2 shows a very strong spatial pattern of record heat waves in a limited set of years in both the observations and GFDL-ESM2G. Given that there should be many ensemble members of GFDL-ESM2G one could test whether this is truly a feature of the model or if this is a statistical anomaly. Secondly, the authors have a nice set of experiments for de-vegetation, but it would be good to have the authors reflect on whether the magnitude of these changes is novel for this particular model if literature exists on such comparisons.

Minor considerations

- 1) Line 57, Something is a bit off here. The sentence reports "some locations experience 22 heat wave days per summer" "...about 2 events per summer ... with the longest heat waves surpassing 10 days in 1934 and 1936." Simple math would suggest that heat wave events would average 11 days given these numbers. I would suggest rephrasing this to avoid such confusion.
- 2) Line 80, What is apparent here is the extent of record-breaking heat,rarity is the geographic extent (not magnitude of actual temperatures). It is likely that the two are related, but perhaps a more precise wording choice here referring to the spatial extent of extremes would help.
- 3) Line 180, Perhaps mention here that 15% is the reference fractional extent of bare soil in this region. How does this compare with real-world conditions? Likewise, while the authors correctly state that the extent of de-vegetation during the Dust Bowl is not well quantified, is there any information that may relate how realistic the different de-vegetation scenarios to what may have happened?
- 4) Line 194, I would consider adding on one of the main figures in the paper (Fig. 2d) the extent of land covers where C3/C4 de-vegetation occurred. I am unsure of whether land cover is distributed across a pixel or whether each pixel has a land cover type. The comment about de-vegetation and higher signal over the forested region has me a bit confused as to whether the de-vegetation experiments added barren land in forested regions or whether the heat-wave impacts in these forested areas are part of a regional signal or advected from de-vegetated crop/grasslands.
- 5) Line 200, The comment on albedo effects on precipitation leads to believe that the authors looked just at the direct effect of albedo on precipitation. I don't know if this matters much, but the indirect effect of aerosol loading on increasing CCN and reducing precipitation efficiency may be another pathway. I don't think this needs to be explored, but references to work could be considered.
- 6) Figure 1: For b and c, please state what time period these refer to; for b I think this is 1930-1937; for c this is the single decade for one model ensemble run.
- 7) Line 553, "Atlantic" , also perhaps express units here of HWF of days/summer.

Reviewer #2 (Remarks to the Author):

The author has fully taken into account my comment, my feeling is that the manuscript is much more readable and has now a format adapted to nature communication. In general the study is very relevant and interesting therefore I would recommend its publication in Nature Communication.

Reviewer #3 (Remarks to the Author):

After carefully reviewing the revised manuscript, the response to my particular comments, and the responses to the other reviewers, I want to commend the authors on their efforts. Not only has the manuscript been improved, but the level of detail, robustness, and politeness the authors employed is admirable. I firmly believe this is a solid and strong contribution to the science and recommend publication.

Reviewers' comments:

Reviewer #1 (Remarks to the Author):

The authors have provided a thorough and thoughtful revision and addressed most of my outstanding questions. I am generally satisfied with this analysis but highlight below a few potential areas to strengthen the results using mainly statistical analyses and integration of results from the literature.

We thank the reviewer again for giving up their time to thoroughly read through our submission and for the suggestions to strengthen our study, particularly around the CMIP5 analysis.

While the authors did look at some attributes from multiple GCMs, their main conclusions seem to come from a single GCM ; this is not necessarily bad, but might benefit from a few additional tests for robustness and, where possible, comparisons to other model simulations. For example, it is unclear whether Figure 1c shows the result from one ensemble member or whether the postulation that GFDL-ESM2G captures decadal peaks in heatwaves is supported using larger ensemble approach.

We acknowledge that our main results stem from a single model, however we do show the mean state results from three separate AGCMs, as well as heat wave time series from the historical runs from 20 CMIP5 GCMs. We have further compared our results to studies that have utilised models with land use changes in the 1930s (e.g., GISS ModelE in the studies by Cook et al.; and the WRF model in Hu et al. 2018), meaning we have made every attempt to compare our results to as many models as possible. With regards to the Reviewer's point about Figure 1c and the single GFDL-ESM2G member, most modelling centers only conducted one long (i.e., multi-century) preindustrial control run, from which the historical runs branched off. That is the main reason why we analysed across multiple centuries and multiple models. We argue that analysing 22 CMIP5 models across 500 years each (except for bcc-csm1-1m which conducted a 400-year long experiment) is sufficient enough to determine the decadal nature of heat waves in unforced runs.

I do appreciate the additional figures provided in the response to reviewers, but it made me further question if the postulation that such peaks in heat waves are really decadal in nature in the GCMs. There could be a short statistical analysis that queries whether the decadal peaks in heat waves as simulated by the models happen at rates more significantly more frequently than if heat events were simulated through some random process without serial correlation.

This is an excellent point that we had not actually considered (i.e., the random nature of clustered events). To test the Reviewer's point about whether decadal peaks in heat waves occur more significantly more frequently than through some random process, we conducted a bootstrapping analysis of the heat wave duration "decadal clusters" for each model across their piControl experiments (referring to the Figure R1 results shown in our previous response document). Here, clusters are defined as 4+ years in a decade. For each model, we count the number of decades with four or more years with significant heat wave durations recorded (i.e., that surpass one standard deviation above the long-term, multi-century mean). Then, with replacement, we bootstrapped the 500-years of heat wave durations for each model 10,000 times to find the median, inter-quartile range and 5-95% confidence intervals of the resampled data. The results, for heat wave duration, are shown below in Figure RR1. We can

see that in a 500-year piControl period, GFDL-ESM2G produces 7 decades of "clusters" with significant heat wave durations (blue dot; see also Figure R1 from the previous response document). The bootstrapped median is four decades, while the 95th percentile is 8 decades, so the result is not quite statistically significant, although it is certainly rare and not very likely to occur by chance. Other models, such as IPSL-CM5A-LR and MPI-ESM models show statistically significant cluster frequencies, above what we'd expect from random sampling. Of the 21 CMIP5 models shown here (leaving out bcc-csm-1-1m with 400 years), 11 models have decadal cluster frequencies above the bootstrapped third quartile. What this analysis shows is that there is a group of models that produce a higher frequency of heat wave clusters in a given decade, above what we'd expect from random sampling. We see similar results for heat wave frequency and amplitude (i.e., the same models show significant frequencies of decadal clusters). Hence, decadal clustering is a feature of a sub-group of CMIP5 models, but not a common feature amongst all models. We now added a paragraph on this in the Methods section of the revised manuscript.

Heatwave duration

Figure RR1: Number of CMIP5 preindustrial control run decades with clusters of 4+ years with extreme heat wave duration summers (blue dots; defined as the one standard deviation above the multi-century mean). For example, for ACCESS1-0, it features 6 decades in 500 years that have 4+ clustered years in those decades with extreme heat wave conditions. The box and whiskers indicate boot-strapped statistics based on 10,000 samples of each model's 500 years, resampled with replacement. The mid-line indicates the median, the box is the interquartile range, and the whiskers are the 5-95th percentiles.

Similarly, Figure R2 shows a very strong spatial pattern of record heat waves in a limited set of years in both the observations and GFDL-ESM2G. Given that there should be many ensemble members of GFDL-ESM2G one could test whether this is truly a feature of the model or if this is a statistical anomaly.

Apologies for not comprehensively explaining what Figure R2 represented. It showed record-breaking decades for heat wave frequency in a selection of ten CMIP5 models and two

observations for records in a single century period, out of a possible five preindustrial control centuries. This is to visually highlight the method we used to determine the record-breaking years shown in Supplementary Table S1. We could have shown all five different versions of this figure for each century for all 22 models, however the reason we showed piControl years 201-300 is that it includes the GFDL-ESM2G model decade that forms part of Figure 1c. Of the models shown, these are in alphabetical order, so we were simply showing models from A to G. The particular decade in GFDL-ESM2G was chosen in Figure 1c because it was most similar in terms of spatial extent to the observed heat wave frequency record. But we want to reiterate, these types of events are rare in the CMIP5 piControl runs. This is clarified in the Methods section.

Secondly, the authors have a nice set of experiments for de-vegetation, but it would be good to have the authors reflect on whether the magnitude of these changes is novel for this particular model if literature exists on such comparisons.

That is a great point. Unfortunately, our experiments are the first of their kind for this particular model, HadGEM3. Therefore, we can only compare our results to studies that have utilised AGCMs to undertake devegetation experiments. This leads to the Dust Bowl studies using GISS ModelE (i.e., papers by Cook et al. 2008/2009). Other studies that have focused on the Dust Bowl drought using AGCMs, such as Hoerling et al. 2009, did not undertake devegetation experiments, which is one reason why they found the drought was not well captured in the northern Great Plains in their SST-only runs using NCAR CCSM3, GFDL AM2.1 and NASA SIPP model. That is why we suggested the magnitude of the changes (in heat wave frequency) in our 50% and 80% soil runs are significant over the Great Plains if they lie outside the range of the 10-member SST-only historical runs.

For example, the summer-mean average surface temperature over the southern Great Plains, averaged over 1930-37 for the 30%, 50% and 80% soil runs from HadGEM3 are 30.7°C, 32.8°C and 35.5°C, respectively. The range of the 10 historical runs over the same period and region is 30.2°C to 31.3°C; this means the surface temperatures (and heat wave metrics) in the 50% and 80% soil runs far exceed the simulated historical range and should be considered significant. We have added this short analysis in the Methods section. As is explained in the paper, we do not have multiple members to test the internal variability in the devegetation experiments. That is why we inserted these two sentences in the Methods: *"As the bare soil sensitivity experiments are rudimentary in their design, the aim was not to realistically quantify their imprint on the Dust Bowl heat waves, but to gauge the impact devegetation can have on the amplification of heat waves and to test for land-atmospheric feedbacks. As with previous studies using land cover estimates from the 1930s^{4,13}, the assumptions of bare soil remain quite subjective."*

Minor considerations

1) Line 57, Something is a bit off here. The sentence reports "some locations experience 22 heat wave days per summer" "...about 2 events per summer ... with the longest heat waves surpassing 10 days in 1934 and 1936." Simple math would suggest that heat wave events would average 11 days given these numbers. I would suggest rephrasing this to avoid such confusion.

Apologies for this confusion. This sentence is reporting on an average across the decade and then detailing the longest events in 1934 and 1936. We can see how this might confuse the reader. In saying this, $HWF \neq HWD \times HWN$, as HWD is the longest heat wave, not the average heat wave length. In the interests of this sentence, we have decided to remove the "2 events per summer" part as this detracts from our main point on the number of heat wave days.

2) Line 80, What is apparent here is the extent of record-breaking heat, rarity is the geographic extent (not magnitude of actual temperatures). It is likely that the two are related, but perhaps a more precise wording choice here referring to the spatial extent of extremes would help.

Agreed. We have now added more specific wording. This is now changed to *"In terms of geographic extent, record-breaking heat wave activity is rare in the piControl simulations. Only one CMIP5 piControl decade, in GFDL-ESM2G (model years 286-296), breaks HWF records over more than 50% of northern and southern-central US, similar to observations (Fig. 1c; Supplementary Table S1)*. We have also decided to add a sentence briefly describing how the method to select simulated record-breaking years and pointing the reader to the Methods for further information.

3) Line 180, Perhaps mention here that 15% is the reference fractional extent of bare soil in this region. How does this compare with real-world conditions? Likewise, while the authors correctly state that the extent of de-vegetation during the Dust Bowl is not well quantified, is there any information that may relate how realistic the different de-vegetation scenarios to what may have happened?

We have now changed this sentence accordingly to *"...the percentage of bare soil averaged over the central US in 1930 was increased from an approximate 15% reference fractional extent to 30%..."*. In response to the reviewer's question about real-world conditions, the Hu et al. 2018 study used different percentages of barren or sparsely vegetated land over their slightly-smaller defined Great Plains area (30°–43°N and 95°–105°W) to force the WRF model. In that study, for the period of 1935-1938, this percentage was set to 10%, although the authors note that in areas that were more badly wind eroded, the soil percentage was set to 20% (less severe erosion), 60% (severe erosion) or 80% (most severe erosion). These numbers are based on land cover reconstructions from the US Geological Survey, and compare well to the 15% reference value. It should be emphasised that the 15% fraction is used in all the CMIP5 historical experiments over this region for the 1930s.

For the second question, our values of 30%, 50% and 80% at the start the 1930s decade therefore best represent the different erosion levels described in Hu et al. 2018 for the Great Plains. The bare soil amounts ascribed to wind erosion that Hu et al. 2018 detail come from a map produced by the US Department of Agriculture Soil Conservation Service in 1954 (<https://www.nrcs.usda.gov/wps/portal/nrcs/detail/national/about/history/?cid=stelprdb1049437>). The map shows the greatest erosion occurred over the Oklahoma panhandle. We have overlaid this erosion map onto a map of the dust aerosol response in the 1930s in the 80% bare soil in HadGEM3 (see Figure RR2 below). This shows that the area of the greatest dust amounts corresponds well with areas that faced the greatest erosion. We now reference the old map in our revised manuscript (like the Hu et al. 2018 study).

Figure RR2: (main map) The mean spring response in mineral dust AOD at 550 nm averaged across 1928-1932, associated with the 80% bare soil simulations. Overlaid is a map of severe wind erosion (taken from <https://www.nrcs.usda.gov/wps/portal/nrcs/detail/national/about/history/?cid=stelprdb1049437>) over northern Texas, southern Colorado and Kansas, centred on the Oklahoma panhandle.

4) Line 194, I would consider adding on one of the main figures in the paper (Fig. 2d) the extent of land covers where C3/C4 de-vegetation occurred. I am unsure of whether land cover is distributed across a pixel or whether each pixel has a land cover type. The comment about de-vegetation and higher signal over the forested region has me a bit confused as to whether the de-vegetation experiments added barren land in forested regions or whether the heat-wave impacts in these forested areas are part of a regional signal or advected from de-vegetated crop/grasslands.

Thanks for the suggestion. Even though the extent is covered in Supplementary Figure S6 for the devegetated soil amounts, we now show the bare soil amounts (as a %) in Figure 2d (see Figure RR3 below). Over the southern region, the bare soil amount in the reference runs (HIST) are 16-18%. Regarding the land cover, for the reviewer's interest, we point them to this link: <http://jules-lsm.github.io/vn4.2/overview.html>, which describes how the land types are treated in the land-surface model: Joint UK Land Environment Simulator (JULES) with sub-grid heterogeneity. Each grid cell (i.e., pixel) in JULES is made up of different fractions of the nine surface types (or tiles): broadleaf trees, needleleaf trees, temperate grass, tropical grass, shrubs, urban, inland water, bare soil and ice. From this, surface fluxes for each tile can be calculated. For our devegetation experiments, we convert the grass fractions of each grid cell to higher bare soil fractions. The remainder of the fractions of each grid cell, whether it be made up of trees or shrubs or both, remains the same.

Figure RR3: As with Figure 2d in the study but overlaid with the reference bare soil amounts (%) at 1930. These amounts are the fractional amounts of bare soil per grid cell.

For the grid cells that are predominantly surfaced with tree cover, these stay predominantly forested, with grid cells in the southern/eastern part of the central US consisting of needleleaf trees, whereas cells further north consist of broadleaf (see Figure RR4 below). These fractions (or percentages) do not change in the devegetation experiments – only grass tiles are converted to bare soil. This means for the 80% soil run, the surface types in the far southeast will be mostly consisting of needleleaf trees (~30-40%) and bare soil (~30-40%; see Figure S6e in the Supplementary Material).

Figure RR4: Percentage of each grid cell in 1930, where the surface is made up of (a) broadleaf trees, (b) needleleaf trees and (c) shrubs.

With respect to the greater heat wave activity in the southern tropical part of the region: this ultimately stems from a combination of lower evaporative fractions and drier soils (as shown in Supplementary Figure S8) which lead to higher temperatures. We argue it is in mostly a regional signal, while a small part (along the southern coastline) is advected from the

devegetated regions. This is highlighted in the horizontal temperature advection plots for the three soil runs, averaged over the summers of 1930-1937 (see Figure RR5 below). Unfortunately, as we did not have access to daily 10-m winds, we had to rely on monthly 1000 mB winds and monthly surface mean temperatures, so there is a degree of uncertainty in these results. Away from the southern coastline, there is only marginal advection ($\sim 90^{\circ}\text{W}$, $\sim 34^{\circ}\text{S}$), however this does not vary greatly between the runs. There is strong advection along the coastline, which appears to match the region of greatest spring heat fluxes (e.g., evaporative fraction). As we detailed in the previous submission, the main reason for the increase in the number of heat wave days in the south (compared to the northern part of the region) is because of the narrower temperature distribution. Also, there is greater heat wave activity in the other model ensembles (HIST , ATL_{HIST} , PAC_{HIST}) in the south compared to the north (referring to Figures 2b-d), so devegetation certainly amplifies this regional signal. As described in the study, this is a common bias in the AGCMs studying the Dust Bowl period. We now briefly mention the temperature advection result in the revised manuscript.

Figure RR5: Horizontal temperature advection (K/day), averaged over JJA for 1930-37 for the (a) 30%, (b) 50% and (c) 80% soil runs from HadGEM3. Temperature advection is calculated using monthly surface temperatures and 1000 mB winds, due to the unavailability of daily 10-m winds.

5) Line 200, The comment on albedo effects on precipitation leads to believe that the authors looked just at the direct effect of albedo on precipitation. I don't know if this matters much, but the indirect effect of aerosol loading on increasing CCN and reducing precipitation efficiency may be another pathway. I don't think this needs to be explored, but references to work could be considered.

This is a very interesting point – one which could lead to an entirely separate study. We can only speculate on whether the dust aerosol loading influenced the precipitation efficiency. The Cook et al. studies used the GISS ModelE where the dust aerosols did not impact cloud microphysics, whereas HadGEM3 does include both the direct and indirect effects (Mulcahy et al. 2018). What we can say, from the best evidence that is available, is that dust aerosols warm the atmosphere, which can reduce the relative humidity and cause cloud droplets to evaporate (if solar radiation is absorbed by the dust particle within the droplet), reducing cloud cover and increasing temperatures further. We have now added a reference in the paper (Lohmann & Feichter 2005) which details the global indirect aerosol effect.

6) Figure 1: For b and c, please state what time period these refer to; for b I think this is 1930-1937; for c this is the single decade for one model ensemble run.

Now added. For the observations, it is 1930-40, centred on 1935 (see Table 1 in the manuscript). For the model, it is piControl Years 286-296, centred on Year 291. We have now inserted these periods into the b) and c) panel titles, noting 'decadal' refers to 11 years in this context, as explained in the Methods section.

7) Line 553, “Atlantic”, also perhaps express units here of HWF of days/summer.

Well spotted, and now fixed. The units have also been modified, as well as in Figure 2.

References:

Hoerling, M., Quan, X.-W., and Eischeid, J. (2009), Distinct causes for two principal U.S. droughts of the 20th century, *Geophys. Res. Lett.*, 36, L19708, doi:10.1029/2009GL039860.

Hu, Q., Torres-Alavez, J. A. & Van Den Broeke, M. S. (2018): Land-cover change and the ‘Dust Bowl’ drought in the U.S. Great Plains. *J. Clim.* 31, 4657–4667.

Lohmann, U. and J. Feichter (2005). Global indirect aerosol effects: a review. *Atmos. Chem. Phys.*, 5, 715–737.

Mulcahy, J. P., Jones, C., Sellar, A., Johnson, B., Boutle, I. A., Jones, A., et al. (2018). Improved aerosol processes and effective radiative forcing in HadGEM3 and UKESM1. *Journal of Advances in Modeling Earth Systems*, 10, 2786–2805.

Reviewer #2 (Remarks to the Author):

The author has fully taken into account my comment, my feeling is that the manuscript is much more readable and has now a format adapted to nature communication. In general the study is very relevant and interesting therefore I would recommend its publication in Nature Communication.

We are grateful to the reviewer for undertaking another review of our manuscript, and for recommending acceptance.

Reviewer #3 (Remarks to the Author):

After carefully reviewing the revised manuscript, the response to my particular comments, and the

responses to the other reviewers, I want to commend the authors on their efforts. Not only has the manuscript been improved, but the level of detail, robustness, and politeness the authors employed is admirable. I firmly believe this is a solid and strong contribution to the science and recommend publication.

Many thanks to the reviewer for giving up their valuable time to read our manuscript again and provide their recommendations.

REVIEWERS' COMMENTS third round:

Reviewer #1 (Remarks to the Author):

The authors have done a marvelous job with their responsive to my comments with their additional analysis and commentary. This is an interesting and useful contribution to the literature.

Response to reviewer

Reviewer #1 (Remarks to the Author):

The authors have done a marvelous job with their responsive to my comments with their additional analysis and commentary. This is an interesting and useful contribution to the literature.

We thank the reviewer for their extremely helpful suggestions throughout the review process and their kind recommendation.